# Vibration hotspots reveal longitudinal funneling of sound-evoked motion in the mammalian cochlea

Nigel P. Cooper [1], Anna Vavakou[1] & Marcel van der Heijden [1]

The micromechanical mechanisms that underpin tuning and dynamic range compression in the mammalian inner ear are fundamental to hearing, but poorly understood. Here, we present new, high-resolution optical measurements that directly map sound-evoked vibrations on to anatomical structures in the intact, living gerbil cochlea. The largest vibrations occur in a tightly delineated hotspot centering near the interface between the Deiters' and outer hair cells. Hotspot vibrations are less sharply tuned, but more nonlinear, than basilar membrane vibrations, and behave non-monotonically (exhibiting hyper-compression) near their characteristic frequency. Amplitude and phase differences between hotspot and basilar membrane responses depend on both frequency and measurement angle, and indicate that hotspot vibrations involve longitudinal motion. We hypothesize that structural coupling between the Deiters' and outer hair cells funnels sound-evoked motion into the hotspot region, under the control of the outer hair cells, to optimize cochlear tuning and compression.

---

[1] Department of Neuroscience, Erasmus MC, Room Ee 1285, P.O. Box 2040, 3000 CA Rotterdam, The Netherlands. Correspondence and requests for materials should be addressed to M.v.d.H. (email: m.vanderheyden@erasmusmc.nl)

**B**efore sound-evoked vibrations are converted into the neural signals that underlie our sense of hearing, the inner ear separates them by frequency and compresses them non-linearly into a physiologically manageable dynamic range[1–5]. Each location along the length of the spiraling cochlear partition is tuned, in a level-dependent manner, to its own characteristic range of frequencies: high frequencies stimulate the cochlear base, and low frequencies, the apex[1]. This place-based spectral analysis, or mechanical tonotopy, underlies the brain's ability to distinguish and identify sounds, even when multiple sound sources are present simultaneously. Exactly how the analysis is achieved is not understood, although the mechanical and physiological properties of key anatomical components have been studied in some detail[1,6–12]. Many cochlear models rely on a hypothetical active process (the so-called "cochlear amplifier"[13,14]) to mimic

contemporary physiological data, but evidence of cycle-by-cycle power amplification in real cochleae is sparse and indirect[15–18], and the underlying assumptions made in many studies are not universally accepted[19–21]. A more complete understanding is currently limited by the difficulty of measuring the sound-evoked vibrations of key structures in intact cochleae. These structures are microscopic in scale and poorly accessible in vivo, and their vibrations are both small (~0.1–100 nm) and physiologically vulnerable[4].

Recent innovations in optical recording techniques, in particular high-speed optical coherence tomography (OCT), have begun to yield important new insights into the cochlea's micro-mechanics. One of the most significant recent findings is that the largest intra-cochlear vibrations do not occur on the basilar membrane (BM; Fig. 1), the structural element of the hearing

**Fig. 1** In vivo imaging and anatomy of the gerbil cochlear partition. **a** Experimental approach to the middle-ear, basilar membrane, and underlying structures of the cochlear partition in the round window region of the gerbil cochlea. **b** Vibration measurements are made by aligning the OCT beam with discrete points spanning the length (red dots) and width (blue dots) of the twisting, tilting, and spiraling cochlear partition (yellow). The measurement technique is sensitive only to vibration components that align with the near-vertical optical axis of each OCT beam. **c** OCT reflectance image (grayscale), with structural framework of Corti's organ (yellow) superimposed for reference (cf. **d**, **e**). Scale bar (red), 0.1 mm. **d** Underlying anatomical structures. The hearing organ's sensory cells are shown in dark blue, with key mechanical support cells and accessory structures in yellow. Thick black lines replicate elements of the main structural framework from **c**: the framework includes the collagen-rich arcuate and pectinate zones of the BM, the tubulin-rich inner and outer pillar cells that form the main (inner, triangular) tunnel of Corti, and the actin-rich reticular lamina (including the cuticular plates of the sensory inner and outer hair cells). **e** Map of vibration magnitudes evoked by a multi-tone stimulus at 70 dB SPL. Magnitudes are expressed on a logarithmic color scale, in decibels (dB) relative to the maximum root-mean-square magnitude of 10 nm. Measurement beam ordinates are indicated at the base of the map using gray arrowheads. b Boettcher's cells, BM_AZ/PZ basilar membrane arcuate and pectinate zones, CAP compound action potential, dc Deiters' cells, ISL inner (primary, osseous) spiral lamina, ihc inner hair cell, iss inner spiral sulcus, ohc outer hair cells, OSL outer (secondary) spiral lamina and ligament, ot outer tunnel of Corti, RM Reissner's membrane, RWM round window membrane, sc support cells, SCC semicircular canal, sm scala media, st scala tympani, sv scala vestibuli, t tectal cells, tc tunnel of Corti (inner). Preparation RG16760, CF 40 kHz

organ that has been the focus of important experimental and theoretical work for over 50 years[4]. Tissues located closer to the sensory inner and outer hair cells (IHC, OHC) of the organ of Corti, including the reticular lamina (RL) and tectorial membrane, have recently been shown to vibrate with much larger amplitudes, different phases[22–29], and sharper tuning[25,26] than the BM.

In the current study, we apply OCT techniques to the basal turn of the gerbil cochlea. Unlike other OCT studies that image the intra-cochlear structures through the cochlea's bony wall, our recordings are made through the transparent round window membrane. The superior spatial resolution permitted by this approach allows us to construct detailed spatial maps of the intra-cochlear vibrations. These maps are inconsistent with the fundamental predictions made by many active models of cochlear mechanics. They reveal a sharply delineated vibration hotspot in the vicinity of the OHC and Deiters' cell bodies that stands out from the surrounding structures in several respects: it has much larger vibration amplitudes, broader frequency tuning, and a hyper-compressive dependence on sound intensity. Abstracts of this work have been presented at recent scientific meetings[30,31].

## Results

**Imaging structure and function through the round window.** Optical measurements of the cochlear partition's structure and mechanical function were made in the basal turns of living gerbil cochleae (Fig. 1; see Methods). The views obtained through the transparent round window membrane yield anatomical and physiological measurements with high spatial resolution: the fluid-filled scalae, sulci, and tunnels are readily differentiated from the cellular and extracellular structures that make up the organ of Corti, and key cellular components can be identified from their positions within the organ (Fig. 1c, d). Observing this level of anatomical detail in intact, living cochleae is unprecedented: even the two layers of collagen fibers that make up the pectinate zone of the BM can be distinguished quite clearly in most images, the highly oriented fibers acting as strong scatterers of the infrared light. The intersection between the arcuate and pectinate zones of the BM is almost always clearly visible, and facilitates the precise registration of a virtual structural framework (Fig. 1c, yellow) that we will use to facilitate interpretation of our results in subsequent figures. This framework delineates the

arcuate and pectinate zones of the BM, the tubulin-rich inner and outer pillar cells, and the actin-rich RL that reaches out laterally to the tectal (support) cell region (Fig. 1d; see ref.[32] for review). The anatomical structures and dimensions that we observe in vivo are consistent with anatomical studies on fresh post-mortem preparations, but differ considerably from those observed in fixed tissue[33]. The physiological measurements of Fig. 1e illustrate that sound-evoked structural motion is focused near the center of the organ of Corti, with distal structures, including the round window membrane, Reissner's membrane, and the limbal zone of the tectorial membrane, vibrating around 3–5× (i.e., 9–15 dB) less than those near the center of the organ. This focusing of the sound-evoked vibration is frequency-dependent, and becomes even more pronounced at lower sound pressure levels, as described below.

**Vibration hotspots.** Figure 2 compares an OCT image of the 23-kHz region of the cochlea with a map of the vibration amplitudes evoked by the characteristic frequency (CF) component of a 40-dB-SPL multi-tone stimulus. The vibration map (Fig. 2c; see Supplementary Note 1 for raw data) shows that the largest vibrations occur in a highly restricted and well-delineated area well within the main body of the organ of Corti. This vibration hotspot centers near the bases of the OHCs and the apices of the adjoining Deiters' cells. It extends depth-wise from the RL to within ~20 μm of the BM, and width-wise from the outer pillar cells to the outer tunnel of Corti (i.e., the interface between the OHC/Deiters' cells and the Hensen's/tectal cell area). We refer to this region as the OHC/Deiters' cell vibration hotspot, and focus the current paper on characterizing its acousto-mechanical properties.

We estimate the dimensions of the OHC/Deiters' cell hotspot to be ~50 μm × ~30 μm at its 10-dB-down points. The hotspot has a sharp boundary, and the vibration amplitudes change only gradually either inside or outside this boundary. Away from the hotspot, vibration amplitudes decrease progressively across the radial extent (i.e., the width) of the cochlear partition, including the BM and supporting cells of the organ of Corti.

We observed sharply focused vibration hotspots in the OHC/Deiters' cell region of all healthy cochleae (n = 31) tested, over wide ranges of sound frequency (Supplementary Note 2, Supplementary Figs. 2–3) and intensity (Fig. 3). Hotspots in the

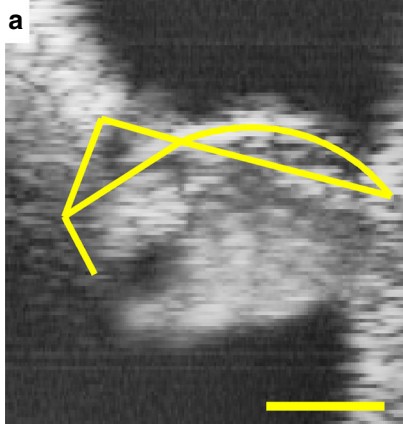
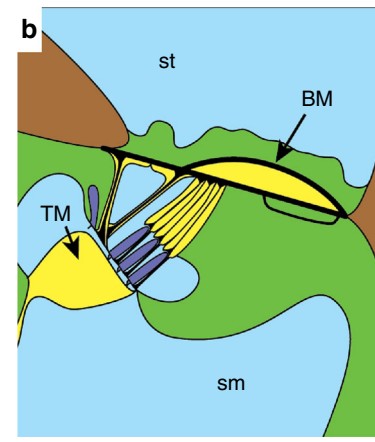
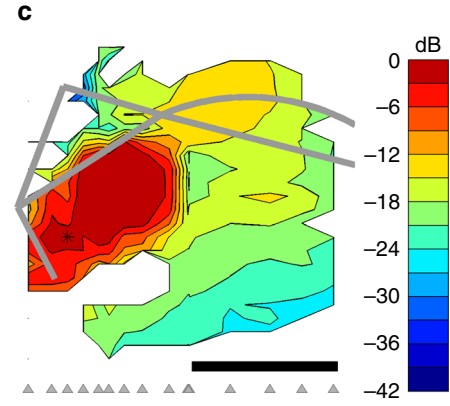

**Fig. 2** A vibration hotspot at low sound levels in the organ of Corti. **a** OCT reflectance image (grayscale), with structural framework of Corti's organ (yellow) superimposed for reference. Scale bar (bottom), 0.05 mm. **b** Underlying anatomical structures, labeled as in Fig. 1c. **c** Map of vibration magnitudes evoked by the 22.6 kHz component of a multi-tone stimulus at 40 dB SPL. Magnitudes are expressed in decibels relative to the 3-nm maximum observed at the location marked with an asterisk. Measurement beam ordinates are indicated at the base of the map using gray arrowheads, and the axial (vertical, z) resolution is ~8 μm. Vibrations clearly focus on a hotspot that encompasses the OHC and Deiters' cell bodies, and sits well within the boundaries of the organ of Corti's structure (gray framework). Scale bar, 0.05 mm. Preparation RG16725, CF 23 kHz

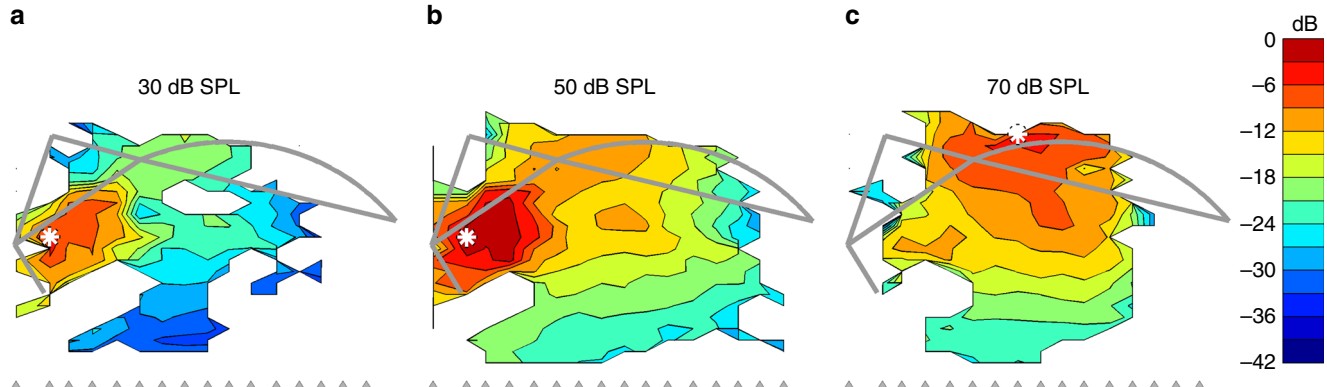

**Fig. 3** Level-dependent modes of vibration near CF. Maps of vibration magnitudes evoked by the 22.6 kHz component of a multi-tone stimulus at three sound pressure levels: **a** 30, **b** 50, and **c** 70 dB SPL. Magnitudes are expressed in decibels relative to the 4.1-nm maximum observed at the location marked with the white asterisk in **b** (asterisks in **a** and **c** mark the sites of peak amplitudes at 30 and 70 dB SPL = 2.6 and 2.1 nm, respectively). Measurement beam ordinates are indicated at the base of each map using gray arrowheads. The structural framework of the organ of Corti is shown for reference in gray (cf. Figs. 1, 2). Data were obtained from the same preparation as in Fig. 2, CF 23 kHz

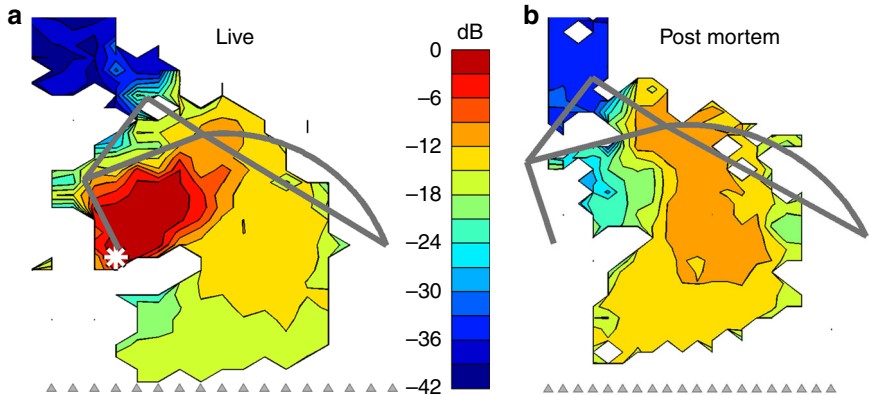

**Fig. 4** Post-mortem disappearance of a vibration hotspot. **a** In vivo vibration map for the 11.2 kHz component of a 65-dB-SPL multi-tone stimulus, showing a clear hotspot in the OHC/Deiters' cell region. **b** Post-mortem vibration map at the same frequency, and evoked by the same stimulus in the same cochlea, within 30 min after death. Vibration magnitudes in both panels are expressed in decibels relative to the 2.6-nm maximum observed in vivo at the location marked with the white asterisk in **a**. The hotspot vibrations reduce at least 10-fold (i.e., 20 dB) post-mortem, while vibrations elsewhere in the body of the organ of Corti increase and BM vibrations remain essentially unchanged. Preparation RG17793, CF 20 kHz

OHC/Deiters' cell region were reduced and/or absent at high sound levels (cf. Figs. 1e, 3c), and occasionally even inverted (i.e., turned into equally-distinct coldspots) for high-frequency and high-intensity stimuli. OHC/Deiters' cell hotspots were also reduced in physiologically compromised cochleae, and disappeared from healthy cochleae soon after death (Fig. 4). Whenever the OHC/Deiters' cell hotspot was reduced, inverted or absent, the organ of Corti's peak vibration amplitudes switched to one of two other locations in a frequency-dependent way: low-frequency (<~5–10 kHz) responses became dominated by vibrations in the lateral (Hensen's) support cell region ~50–100 μm away from the OHC/Deiters' cell area, while high-frequency responses became dominated by BM vibrations (Fig. 3c).

**Broader frequency tuning of the hotspot.** Mechanical tuning of BM vibrations is remarkably similar (although not entirely identical) to electrophysiological tuning in IHCs, OHCs, and the auditory nerve[3,4,34–36]. As the OHC/Deiters' cell vibration hotspot is sandwiched between the BM and the site of stimulus transduction by the IHCs and OHCs, one might expect the frequency selectivity in the hotspot to approximate IHC, OHC, and neural tuning at least as well as the BM does. To test this hypothesis, we compared the tuning properties of three key

locations in and around a vibration hotspot (Fig. 5). While all three of these locations are tuned to similar frequencies (~23 kHz, Fig. 5a) and have similar overall phase characteristics (Fig. 5b), there are major differences between the overall shapes and/or depths of their tuning curves. The BM is the most sharply tuned of the three sites in terms of its tip-to-tail ratio (showing a 45-dB difference in gain from ~0.4 to 23 kHz), followed by the heads of the Deiters' cells in the hotspot itself (tip–tail ratio ~31 dB) and the tectal/Hensen's (supporting) cell region ~50 μm lateral to the hotspot (tip–tail ratio ~20 dB). Although a direct comparison with auditory nerve data from this region of the gerbil cochlea is difficult, the BM's tuning is the closest to matching typical high-frequency neural tuning curves, whose tip–tail ratios exceed 40 dB[34,37].

Other differences between the tuning curves in Fig. 5 are also noteworthy. The heads of the Deiters' cells vibrate ~3× more than the BM around CF, and >10× more than the BM at low frequencies. Hensen's and tectal cells vibrate ~2–3× less than the BM near CF, but 3–10× more than the BM below 5 kHz. Deiters' cell vibrations phase-lead BM vibrations by ~0.1 cycles around CF and up to 0.4 cycles at low frequencies, while Hensen's/tectal cell vibrations phase-lead BM vibrations by ~0.1 and 0.3 cycles at low and high frequencies, respectively (Fig. 5b). These systematic differences in tuning between different sites within the organ of

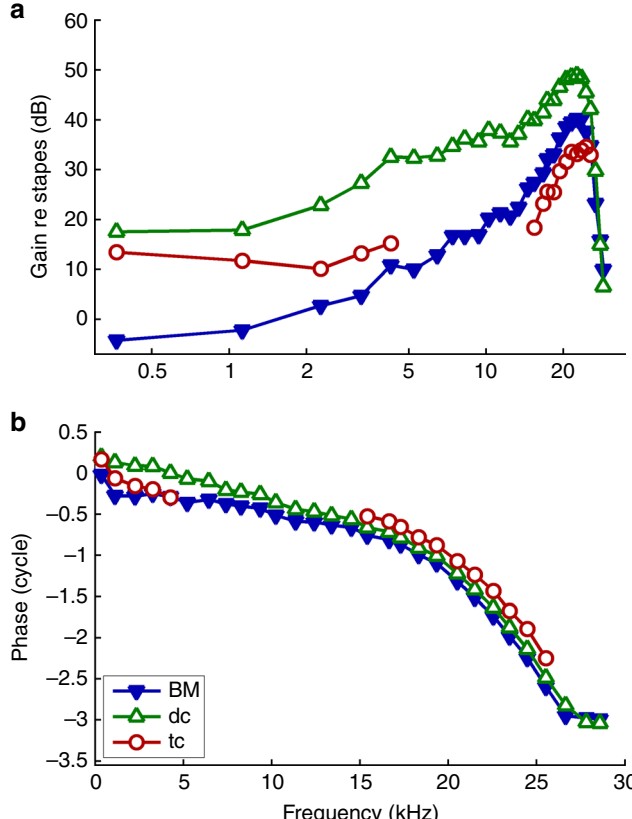

**Fig. 5** Tuning characteristics of the vibration hotspot and adjacent structures. **a** Amplitude- and **b** phase-transfer functions evaluated at three locations in the living organ of Corti: filled triangles, near the junction between the arcuate and pectinate zones of the BM; open triangles, in the Deiters' cell (dc) bodies near the center of the vibration hotspot, ~40 μm away from the BM; and circles, in the Hensen's/tectal cell (tc) region ~50 μm lateral to the Deiters' cells. Gain and phase are referenced to the stapes vibrations evoked by identical multi-tone stimuli. Preparation RG17805, CF 25 kHz (see Fig. 6). Stimulus level 50 dB SPL

Corti demonstrate that the organ's main vibration modes are frequency-dependent (further illustrated in Figs. 6, 7, and Supplementary Figs. 2, 3). As shown explicitly in Fig. 3 and implied in Fig. 6, some of these vibration modes are intensity-dependent, too.

**Hyper-compression in the hotspot**. BM vibrations grow less than proportionally with sound pressure near CF[2–5]. The functional correlate of this nonlinear behavior is a compression of the huge dynamic range of everyday sounds to fit the much smaller dynamic range of the IHC transduction channels. As the OHC/Deiters' cell hotspot is sandwiched between the BM and the IHC and OHC transduction sites (and contains the very OHCs that are thought to mediate compression), motion in the hotspot region is expected to show compressive behavior too.

To compare the nonlinear compression between BM and hotspot we constructed tuning curves, normalized to middle ear motion, for a wide range of sound intensities (Fig. 6a, b). At low frequencies (<0.7× CF) the normalized BM curves coincide (Fig. 6a), indicating linear growth of BM vibration with sound pressure. Only near CF do the BM responses show compressive nonlinearity, visible in Fig. 6a as a gain reduction with increasing intensity. This spectrally selective form of compression agrees with a large body of historical BM data[2–4]. The hotspot vibrations (Fig. 6b) also show compressive nonlinearity, but this differs from

the BM in two major respects: the frequency range and the strength of the compression. Hotspot motion is compressive over the entire frequency range, down to the lowest frequencies tested (6 octaves below CF). Near CF, where both BM and hotspot are nonlinear, the compression is stronger in the hotspot than on the BM: between 20 and 73 dB SPL, the sensitivity at CF drops by 57 dB in the hotspot versus 35 dB on the BM (Fig. 6a, b).

The stark contrast in nonlinear behavior between BM and hotspot is further illustrated by input/output (I/O) functions (Fig. 6c–f). Well below CF, the BM exhibits linear (1-dB/dB) growth (Fig. 6c), whereas all of the low-frequency I/O functions for the hotspot (Fig. 6d) are compressive (<1 dB/dB). The hotspot tuning functions (Fig. 6b) in this frequency range run parallel, indicating that the hotspot nonlinearity behaves identically for all frequencies up to ~17 kHz, i.e., 1/2 octave below CF. When increasing the frequency toward CF (25 kHz) and beyond, even stronger differences between BM and hotspot are revealed. The BM I/O functions become increasingly shallow at higher frequencies (Fig. 6e), but their growth rates remain positive. In contrast, the near-CF hotspot I/O functions exhibit systematic hyper-compression: following their initial growth at low intensities, the hotspot responses start to decrease beyond a certain stimulus intensity (Fig. 6f). The reversal point shifts to lower intensities for the higher frequencies.

**Spatial dissection of hotspot motion**. To analyze the spatial profile of the organ of Corti's vibrations and to further assess its frequency dependence, we defined a quasi-transverse pathway through the cochlear partition (Fig. 7a, red arrow). This path starts at the BM, follows a straight line that runs close to the main axes of the Deiters' cells and OHCs, and ends at the RL. Using the BM as reference for the motion across this pathway, we obtained spatial profiles (one per frequency) of the vibrations within the organ of Corti. These profiles show rapid, frequency-dependent spatial gradients of vibration amplitude (Fig. 7b) and phase (Fig. 7c) over extremely short distances—particularly at low frequencies. In this example, the steepest transitions occur within a 10-μm stretch starting ~10 μm beneath the level of the BM, where vibration levels increase by 15–20 dB (i.e., 5–10×) and phase-leads of over 0.25 cycles accumulate at low frequencies (a 0.25-cycle phase-lead means that hotspot velocity synchronizes with BM displacement).

Beyond the ~10-μm-thick transition layer which forms one edge of the hotspot, vibration amplitude grows more gently and tends to plateau, typically peaking slightly around the middle of the OHC region. Beyond these peaks, the amplitudes fall by 2–3 dB (i.e., to 0.7–0.8×) towards the RL and tectorial membrane (at the OHCs' apices). Response phases (Fig. 7c) remain stable across most of the Deiters' and OHC region, particularly at low frequencies. Thus, most of the hotspot moves together, as a single unit (with a deformable framework, e.g., as illustrated in Fig. 7e, f). From a mechanical perspective (see Discussion), it is important that the amplitude and phase variations (from outside to inside the hotspot) are graded with frequency: the largest spatial gradients occur at low frequencies, and the smallest ones around CF (Figs. 5, 7, 8).

The relative motion of any pair of points on a pathway is readily determined by computing the vector difference of their vibrations. In the 50-dB-SPL data of Fig. 7, the differential motion between the two ends of the Deiters' cells soma is 2.0 ± 0.8 nm (mean ± standard deviation, n = 26 frequencies), while for OHCs the end-to-end motion is −0.9 ± 0.5 nm. Assuming that both the pathway illustrated in Fig. 7 and the cellular vibrations that we measure are closely aligned with the principal axis or axes of the OHCs and Deiters' cells, the ratio of these two differential

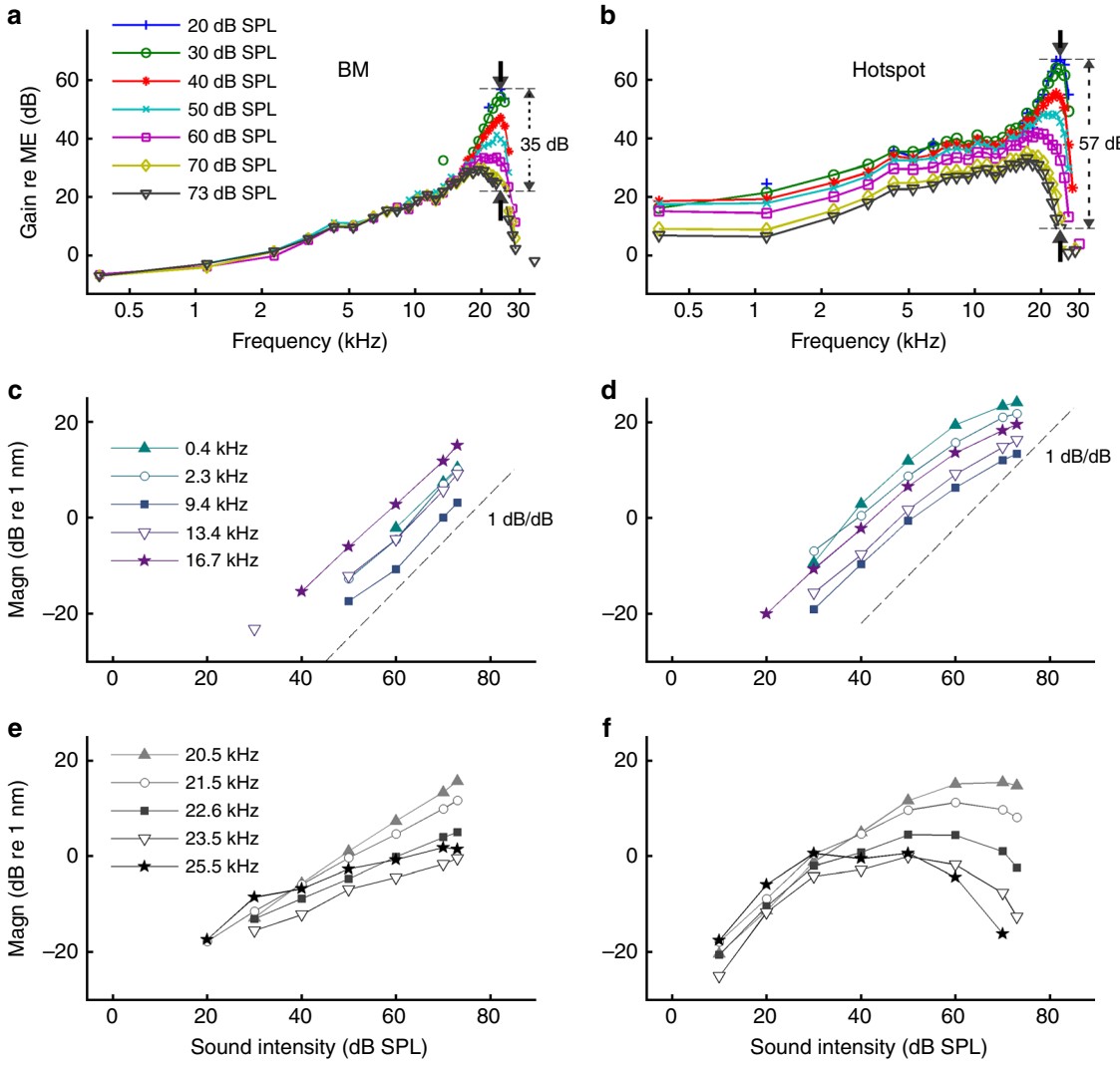

**Fig. 6** Compression on the BM (left) versus hyper-compression in the hotspot region (right). **a**, **b** Vibration magnitude as a function of frequency, normalized to stapes motion. Sound intensity is indicated in the key, with arrows indicating amounts of compression observed at CF (25 kHz) between 20 and 73 dB SPL. **c**, **d** Below-CF input–output functions for different frequencies, based on the tuning curves of panels **a**, **b**. Dashed lines illustrate linear growth for reference. **e**, **f** Near-CF input–output functions. Data were obtained from a single preparation (the same one as in Fig. 5, CF 25 kHz), and acquisition from the two locations was interleaved

motions should indicate the relative length changes of the two cell types. Under these one-dimensional assumptions, the data of Fig. 7 indicate that Deiters' cells change length (cyclically) during sound stimulation by ~2× as much as OHCs do.

The suggestion that Deiters' cells undergo cycle-by-cycle length changes that are greater than the compliment of the OHCs' length changes is difficult to reconcile with prevailing ideas in cochlear mechanics (see Discussion), and there are many reasons to doubt that sound-evoked motion in the cochlea should be limited to the transverse and/or radial dimensions, as assumed in the one-dimensional analysis above. A more plausible alternative interpretation may be that motion in all three dimensions is involved, and that the longitudinal component of this motion becomes particularly significant, especially at low frequencies, in the hotspot region (see Supplementary Notes 3, 4 for the rationale behind this suggestion). It is this longitudinally directed form of motion that is illustrated in Fig. 7d–f and Supplementary Movie 1, and that is tested for experimentally below.

**Testing for longitudinal motion in the hotspot**. To test for the existence of longitudinal motion in our recordings, we studied the

effects of changing the angle of incidence of our measurement beam (Fig. 8). Knowing that the underlying BM's motion is almost entirely transverse in nature (Fig. 8a; see refs.[1,38,39]), the simple geometrical considerations of Fig. 8b predict that any longitudinal motion in the nearby fluid or structures should be sensed with opposite polarities from the two sides of the cochlea's transverse plane (Fig. 8b, c; see Supplementary Note 4 for details).

In most of our experiments, the longitudinal viewing angle of our measurements was such that the incident light-beam pointed towards the apex of the cochlea (cf. $\alpha_1$ in Fig. 8b). In all of these experiments, motion recorded from the hotspot region phase-led the BM in a frequency-dependent manner (cf. Figs. 5, 7, Supplementary Fig. 3). In two particularly revealing experiments, however, we managed to realize a viewing angle that was flipped longitudinally (cf. $\alpha_2$ in Fig. 8b). In these two experiments, the hotspot region was found to phase-lag the BM in a frequency-dependent manner, with the largest phase-differences and amplitude-gains observed at low frequencies (Fig. 8e, f).

This effect of viewing angle on the polarity (lead versus lag) of the relative hotspot/BM phase confirms two of six predicted measurable effects of longitudinal motion derived in

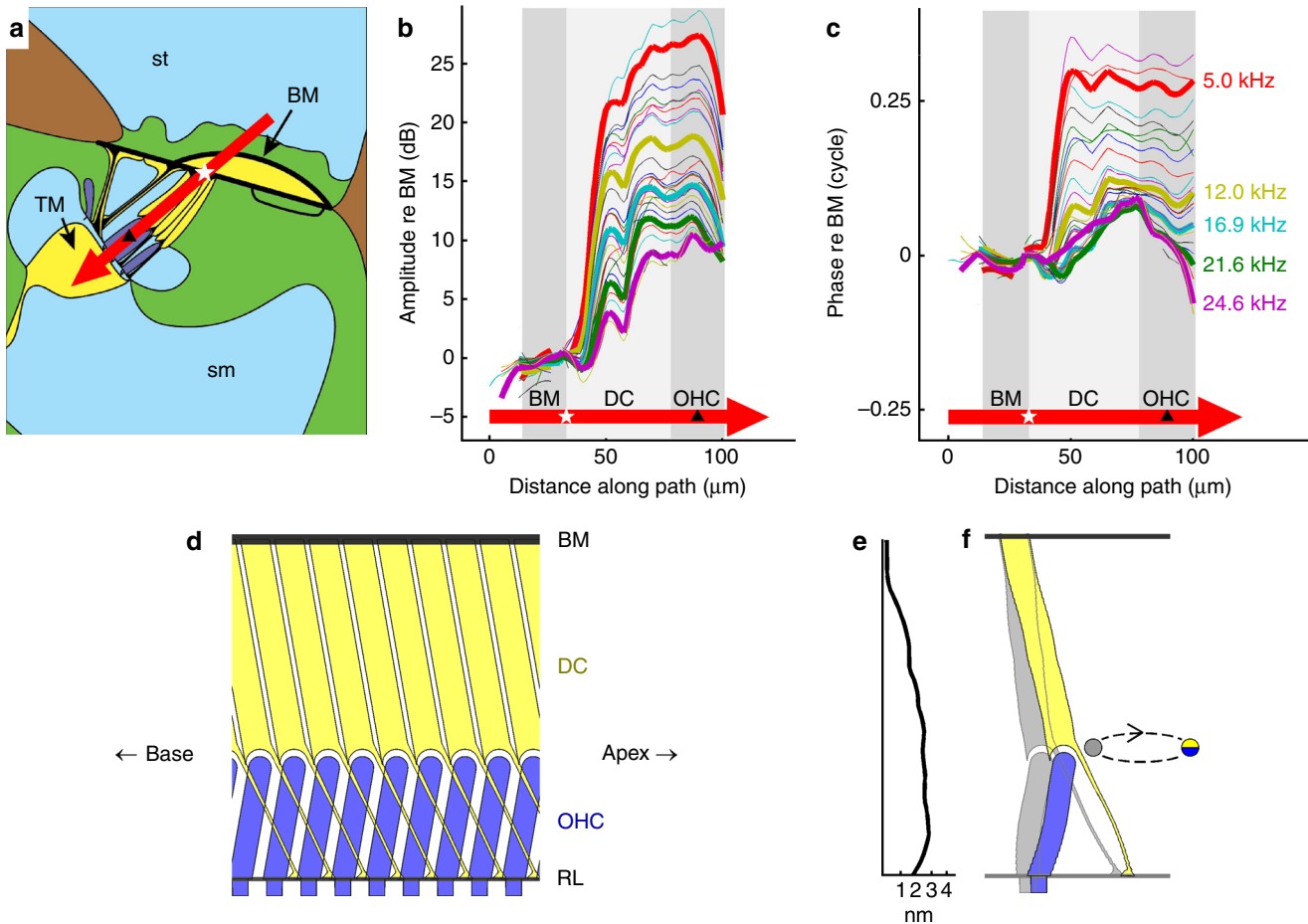

**Fig. 7** Frequency-dependent spatial profiles of a vibration hotspot. **a** Transverse anatomical schematic for the 23 kHz region of the gerbil cochlea (cf. Fig. 2a), illustrating the spatial pathway (red arrow) across which vibration amplitude (**b**) and phase (**c**) were analyzed. The white star marks the reference point on the BM; the black triangle the middle of the outer hair cell region. **b**, **c** Spatial profiles of vibration amplitude and phase for 21 different frequency components (color-coded) in a multi-tone stimulus. The curves for five frequencies (indicated in **c**) are highlighted by increased line width. Amplitudes and phases are expressed relative to those observed on the BM (cf. white star in **a**). Gray patches indicate the different regions traversed by the red arrow, including the BM, Deiters' cells (DC), and OHCs. **d** Schematized anatomical relationship between Deiters' cells (yellow) and OHCs (blue) along the longitudinal axis of the cochlea, perpendicular to the section shown in **a**. The cells are shown suspended between BM and RL, with the OHC stereocilia protruding downwards (beneath the level of the RL). The apical-ward projection of the Deiters' cells soma and phalangeal processes contrasts with the basal-ward projection of the OHC[43, 51]. **e** Measured vibration profile for the 5-kHz data of **b**, recast onto linear coordinates, for comparison with the cartoon in **f**. **f** A hypothetical interpretation of the profile from **e**, based on the idea that motion in the OHC/Deiters' cell hotspot is primarily oriented in the longitudinal direction (see text). Two snapshots of a single OHC/Deiters' cell "unit" are shown, one in gray and the other in colors (blue, OHC; yellow, DC). The snapshots depict the unit at the two extremes of an elliptical vibration cycle. The illustrated movement is highly exaggerated: the measured 2.8-nm maximum displacement is scaled up 1000-fold, relative to the scale of the cellular anatomy, to make it visible in the illustration. Data were obtained from the same preparation as in Figs. 2, 3, CF 23 kHz. Stimulus level 50 dB SPL

Supplementary Note 4. The other four predicted effects were also observed without exception: both phase- and amplitude-differences between hotspot and BM motion were larger at lower frequencies; low-frequency hotspot-to-BM phase-differences often exceeded 0.25 cycles, and occasionally approached 0.5 cycles; and significant hotspot-to-BM phase-differences were still observed from viewing angles that were almost perpendicular to the BM. These observations provide strong support for the idea that vibrations in the OHC/Deiters' cell hotspot involve longitudinal (in addition to transverse, and probably radial) motion, as schematized in Supplementary Movie 1.

## Discussion

The high-resolution vibration maps presented here reveal a well-delineated vibration hotspot whose epicenter lies partway between the RL and the BM. This hotspot extends as far as the

RL, but it also contains both the basal poles of the OHCs and the heads of the adjoining Deiters' cells. This is a significant new finding, building on numerous studies which have already demonstrated that different structures within the cochlear partition move to differing degrees, in different directions, with different phases and different frequency-dependencies[1,22–29]. These findings shed new light on the internal workings of the hearing organ, and are likely to pre-sage significant improvements in the understanding of a very complex system.

The sound-evoked vibration characteristics that we observe in the OHC/Deiters' cell hotspot are strikingly similar to those reported recently at the RL level by Ren et al.[28,29] and by Oghalai's group[24,26]. Ren et al. made sequential measurements from individual loci on the RL and BM, as viewed through the transparent round window membrane in the basal turn of the cochlea, whereas Oghalai's group made more panoramic, simultaneous measurements by applying the OCT technique through

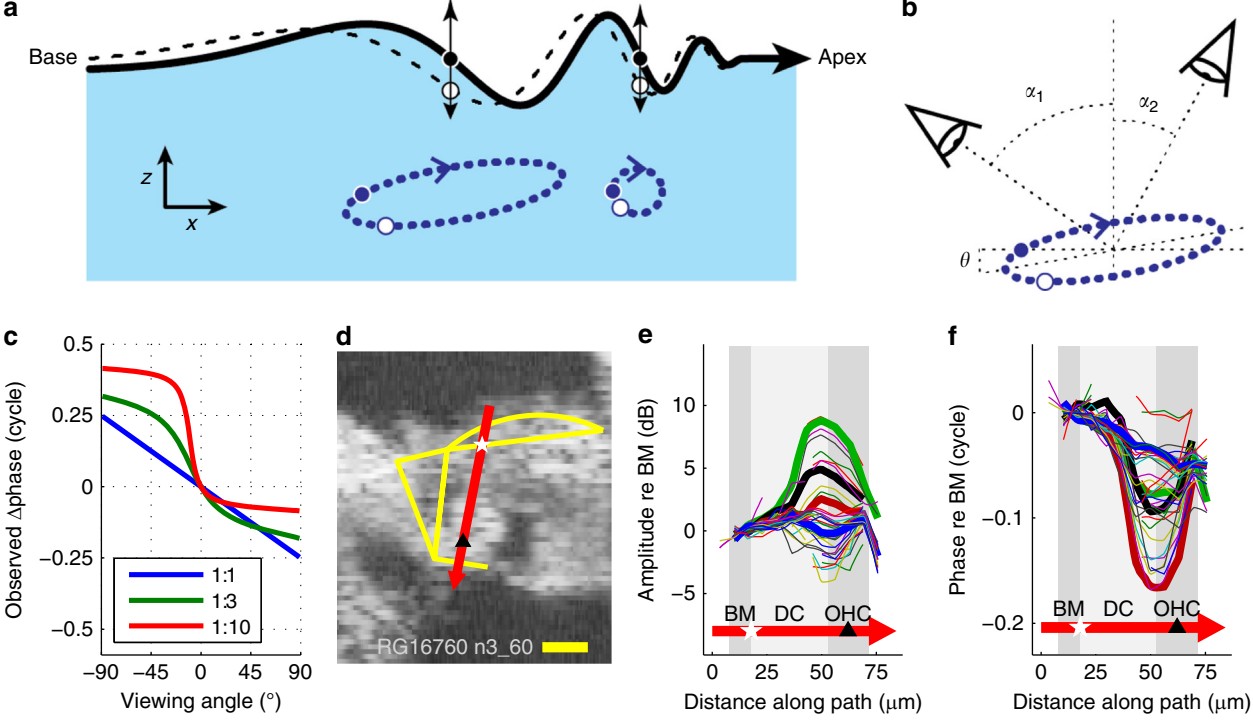

**Fig. 8** Viewing-angle dependent 'relative-phase inversion' of hotspot motion. **a** Contrast between the purely transverse motion of the BM (vertical arrows) and the elliptical trajectories predicted for fluid (and potentially structural) particles underlying a surface wave in the cochlea[1, 38, 39, 68]. The wave's surface (the BM, black line) is pictured at two instants in time, separated by 1/8 cycle, with neighboring fluid particle trajectories shown (using dashed blue lines) at two longitudinal positions. Coordinates: $x =$ longitudinal; $z =$ transverse. Longer wavelengths correspond to more elongated ellipses. The illustrated tilt of the ellipse ($\theta$ in **b**) may result from dissipation in either the fluids[39, 68] or structures within the organ of Corti (see Supplementary Note 3). **b**, **c** Predicted effects of viewing angle on measured motion. In **b**, a "negative" angle $\alpha_1$ views the wave traveling away from the measurement beam, with the measured (projected) motion of the ellipse phase-leading the transverse motion of the BM. By contrast, a "positive" viewing angle $\alpha_2$ observes the same wave as traveling towards the beam, such that the measured motion of the ellipse phase-lags the transverse BM motion. **c** Exemplar relationships between viewing angle and the relative phase of the projected motion for ellipses with 10-degrees of baseline tilt ($\theta$, see **b**) and longitudinal/transverse elongation factors of 1, 3, and 10 (see key). Computations, see Supplementary Note 4. **d** OCT image from one of two preparations where vibration measurements were made from positive angles of incidence ($\alpha_2$ in **b**). Scale bar, 0.025 mm. **e** Profiles of vibration amplitude across the path shown by the red arrow in **d**. Each curve represents one frequency component in the multi-tone stimulus. Four specific frequencies are identified by increased line width: 5.7 kHz (green), 10.6 kHz (black), 17.2 kHz (red), and 37.2 kHz (blue). **f** Corresponding phase data. All profiles are referenced to the vibrations observed at the BM location indicated with a white star in **d**. Gray patches indicate the different regions traversed by the red arrow, including the BM, Deiters' cells (DC) and OHCs (marked by the black triangle). Preparation RG16760, CF 40 kHz. Stimulus level 60 dB SPL

the bony wall of the apical cochlea. Both techniques reveal components of RL motion that are more broadly tuned, more extensively nonlinear, and up to 10× larger than BM motion. Additionally, Ren et al.[28,29] report physiologically vulnerable low-frequency phase leads approaching 0.5 cycles between RL and BM, while Oghalai's group emphasize a much less-vulnerable 0.5-cycle phase difference between the BM and the lateral compartment of the organ of Corti at high frequencies[24]. All of these findings are consistent with observations made in the present report, which combine the panoramic OCT technique with the improved optical access afforded by the round-window approach. The new information added by our study is that: (i) the entire OHC/Deiters' cell hotspot region shares many of the characteristics previously attributed to the RL; (ii) hotspot vibrations exhibit hyper-compression near their CF; and (iii) hotspot vibrations appear to involve longitudinal motion.

Sound-induced vibrations within the hotspot contrast sharply with those of nearby structures like the BM, tectal, and Hensen's cells. The steep amplitude and phase gradients observed only 10–20 μm away from the BM (Figs. 2, 7) underscore the importance of good spatial resolution. Without this, BM, hotspot and other vibrations cannot be isolated with confidence[30]. It is reassuring that our BM data are consistent with the extensively

documented BM measurements obtained with traditional techniques[4], expressing large tip-to-tail ratios, linearity up to 1/2 octave below CF, and shallow but positive growth near CF. The hotspot region behaves very differently from the BM, however, and the challenge now is to identify its underlying mechanism from both physical and functional perspectives.

From a physical perspective, it is highly unlikely that the hotspot's spatial profile (as illustrated in Figs. 2–4, 7, 8) can be underpinned by purely transverse motion within the organ of Corti. This would involve enormous squeezing of the thin fluid-filled layers that make up the transition regions near the feet of the Deiters' cells (at one edge of the hotspot), and would not explain why phase differences of over 0.25 cycles occur between nearby layers at low frequencies. These characteristics are much easier to understand when assuming that the differences in measured magnitude and phase coincide with differences in the direction of motion across nearby layers, i.e., if the layers would be sliding with respect to each other in the radial and/or longitudinal direction. Such motion, with both radially and longitudinally directed components, has been observed directly in at least two previous studies: one using high sound levels in passive cochleae[1], and one using electrical stimulation in isolated turns of excised cochleae[40]. The results of our study (Figs. 7, 8) provide

strong evidence that longitudinal motion is also evoked by low-level sounds in vivo—i.e., in active, functioning cochleae.

Longitudinal motion (which is not to be confused with longitudinal coupling of transverse motion) has rarely been invoked in the analysis of traditional cochlear-mechanical (BM) data. This is partly because the BM is known not to undergo significant longitudinal motion[1]. The ribbon-like BM is firmly clamped at both edges and contains dense layers of stiff, strongly inter-connected fibers[1,32]. It is also sandwiched between two fluid-filled ducts which act anti-symmetrically on its two faces[1,38], providing little net drag in the longitudinal direction. The situation is very different inside the OHC/Deiters' cell hotspot, where motion is likely to be subject to constraints intermediate to those on the BM and in the surrounding fluid (see Supplementary Notes 3, 4). Deiters' cells and OHCs are suspended (or sandwiched) quite freely between the BM and RL[1,32,40,41], and various observations suggest that the RL is much less stiff and less securely tethered than the BM[1,8,41,42]. Deiters' cells, OHCs, and even the RL may therefore offer little opposition to movement in either long-itudinal or radial directions[1,41,43,44].

From a functional perspective, it may be tempting to assign the much larger vibrations seen in the hotspot region a greater relevance to hearing than the smaller and seemingly more per-ipheral vibrations of the BM. Anatomically, the hotspot is clearly closer than the BM is to the transduction sites of the IHC and OHC stereocilia. However, the hotspot's poorer frequency selec-tivity and non-monotonic I/O functions (Figs. 5, 6) contradict a closer (than the BM) functional relationship: auditory nerve tuning is at least as sharp as BM tuning[34,36], and strongly non-monotonic I/O functions of the type illustrated in Fig. 6 are rarely seen in IHC, OHC, or neural recordings[45,46]. (The few non-monotonic features that are observed in recordings from the auditory nerve[47,48] and OHCs[35] are not restricted to CF, and are rarely as strong as the mechanical nonmonotonicities of Fig. 6.) The mere location of the hotspot does not define its function, however, and it is entirely feasible that the BM is closer to the IHCs from a functional perspective than the hotspot region is. It has been known for over 100 years[41] that the triangle of Corti can act as rotating wedge[49] to couple transverse BM vibrations to radial motions in the sub-tectorial space.

The proposed longitudinal nature of the hotspot vibrations may explain why the hotspot's response characteristics are not passed on to the IHCs and auditory nerve. IHC stereocilia are arranged in rows which run almost parallel to the cochlea's spiraling longitudinal axis[32,44]. This makes them highly sensitive to radial deflections, however these might arise[41,50], but insensi-tive to longitudinal shear. In stark contrast, OHCs have char-acteristically V- or W-shaped stereociliary bundles[44,51] which may well be excited by longitudinal motion. Each wing of the OHC's bundle is oriented at a considerable angle to the cochlear spiral, and may therefore act independently to sense and rectify a longitudinally-directed stimulus (e.g., as illustrated in Fig. 1c of ref.[52]). OHC electromotility[12,43] may also induce longitudinal motion in the cochlea, both at the level of the RL (e.g., as illu-strated in Fig. 1c of ref.[43]) and near the OHCs' junctions with the Deiters' cells[40].

The very existence of a frequency-, intensity-, and physiologically-dependent vibration hotspot within the organ of Corti suggests that the classical way of modeling cochlear mechanics, with the BM as the dominant elastic structure and the surrounding fluid as amorphous mass load[39,53] is incomplete. The anatomical structures within the organ of Corti clearly impose stringent constraints on the motion. These constraints require physiological integrity (the hotspot rapidly disappears post-mortem), suggesting a connection with physiologically active processes such as OHC motility[43]. However, the hotspot pattern

is not restricted to the narrow frequency band around CF that is generally associated with active feedback or "cochlear amplification"[12,13,15]; hotspots are observed over the entire fre-quency range, down to at least 6 octaves below CF (Figs. 5, 6; Supplementary Fig. 2). Both the wideband character of the hot-spot's motion and its wideband nonlinearity contrast with the fundamental feature of active models that "the region of activity is spatially limited"[15].

A common idea about OHC length changes is that they serve to "enhance the peak BM response"[54]. This is difficult to reconcile with the observation that BM motion is actually smaller than hotspot motion, and the large (>0.25-cycle) phase-differences between hotspot and BM motion at low frequencies are incon-sistent with the high speed of OHC length changes[12,54]. Active micromechanical models typically describe Deiters' cells as rigid rods that transfer OHC length changes to drive BM motion directly (e.g., ref.[55]), thus predicting zero differential motion between the two ends of the Deiters' cell soma. Our measure-ments show that the end-to-end motion of Deiters' cells is in fact very large, exceeding the OHCs end-to-end motion at least twofold across a wide range of frequencies (Fig. 7). Our mea-surements also fail to reveal the anti-phasic motion (or half-cycle phase shifts) between the top and bottom surfaces of the OHCs commonly predicted by active micromechanical cochlear models (e.g., refs.[54–56]). We see two potential explanations for this: either anti-phasic motion is not excited in healthy, acoustically-stimulated cochleae (perhaps owing to the hearing organ's visco-elastic properties, cf. refs.[8,57]), or it is so small that it becomes swamped by other forms of motion. Presently, we are unable to discriminate between these possibilities.

In the face of the above problems and inconsistencies, we propose an alternative hypothesis, namely that the hotspot motion results from structural constraints that require physiolo-gical integrity, and that OHCs act (on a slower timescale) as mechanical regulators[21,58], rather than as rapid, direct sources of BM vibrations. Supporting evidence for such parametric control comes from temporal sluggishness in cochlear compression[59]. As shown below, a regulatory role of OHCs also helps explaining the peculiar nonlinear behavior of the hotspot (Fig. 6).

We propose that hotspot motion plays an important functional role in enhancing cochlear frequency selectivity. Deiters' cells and OHCs provide an internal scaffolding of the organ of Corti that strongly constrains its possible mechanical deformations under the influence of acoustic stimulation. In a healthy cochlea, this scaffolding funnels the longitudinal motion associated with the BM's traveling wave, causing it to become more tightly focused within the cross-section of the organ of Corti than it would be otherwise be (e.g., in the dead organ, cf. Fig. 4b). When the wave approaches its CF region, its wavelength decreases rapidly, and a second vibration mode emerges that involves larger transverse motion, and hence more BM displacement. Such a rapid change of vibration mode occurs in graded fluid waveguides with strong internal stiffness, and can enhance frequency selectivity without the need for cycle-by-cycle amplification[60]. In this scenario, the post-mortem decline in the integrity of the internal scaffolding explains the simultaneous loss of frequency selectivity and the hotspot pattern.

The nonlinearity of the hotspot motion differs from the well-known BM nonlinearity[2–4] in both its bandwidth and its strength (Fig. 6). Wideband mechanical compression similar to that seen in the hotspot has been observed previously near the apex of the cochlea in chinchillas, guinea-pigs, and mice, using traditional as well as OCT techniques[26,61,62]. These findings have been interpreted to expose qualitative differences between low-frequency hearing (at the apex) and high-frequency hearing (at the base)[4,61,62]. However, basal turn data in the present study and elsewhere[28,29] show that

wideband compression is not unique to the apex. The actual dichotomy may therefore be that between the BM and the organ of Corti.

The wideband character of nonlinearity within the organ of Corti may also have a major impact on the interpretation of otoacoustic emissions (sounds of intra-cochlear origin recorded in the ear canal). Decades of BM data have shaped the widely accepted view that cochlear nonlinearities are confined to a narrow spatial region near CF, and this assumption forms the backbone of nearly all models of distortion product generation. However, there is considerable indirect evidence that some emissions originate more remotely from the CF region, at more basal cochlear sites[63]. The wideband nonlinearity of the hotspot provides further evidence that nonlinear responses to tones are not restricted to the CF site at all, and actually occur over an extended cochlear region spanning multiple octaves. For decades, this phenomenon has been hiding just 20 μm beyond the BM. Its discovery may help to resolve the apparent contradiction between the otoacoustic[63] and BM data[4], and could call for a major revision of models of distortion products in the cochlea.

The wideband compression observed in the hotspot is puzzling from a functional point of view. At the 25-kHz place, frequencies as low as 400 Hz show compressive growth in the OHC/Deiters' cell hotspot (Fig. 6b). Attempts to interpret such compression in terms of amplification (e.g., ref.[28]) are problematic. Low-frequency hearing sensitivity depends only on physiological integrity in apical cochlear structures and is unaffected by the basal OHCs involved in the wideband compression in Fig. 6. Even stranger is the fact that the same 400-Hz component that grows compressively in the hotspot at the 25-kHz place, becomes perfectly linear when observed on the BM at more apical locations. Regardless of whether compression originates from saturating amplification[64] or controlled friction[21], it should cause the entire wave to accumulate nonlinearity during its travel, excluding such a return to linearity.

We therefore postulate that the linear BM motion represents the true carrier of the low-frequency wave, and the hotspot motion merely feeds off this low-frequency, linear BM wave without affecting it in return. The compressive growth evoked by low-frequency sounds in the hotspot (Fig. 6d) then informs us that the hotspot's coupling to the BM is not constant, but becomes weaker with growing intensity. This variable coupling is consistent with the proposed regulatory role of OHCs. Specifically, we propose that the parametric impedance control is performed by the OHCs, which are known to change their axial stiffness tonically when excited[11]. Such a proposal is by no means new (see refs.[11,58]), but has been overshadowed for many years by the competing hypothesis that a "cochlear amplifier" injects extrinsic energy into the BM's traveling wave on a cycle-by-cycle basis. The functional role of a parametric impedance regulation is to compress the inputs to local IHCs, rather than to amplify the BM's motion, and the functional part of this compression is restricted to a narrow frequency band around CF (as reflected by the BM's nonlinearity). But even if the impedance regulation serves to compress only the near-CF input to nearby IHCs, it may still affect other responses to non-CF components—especially the vibrations of the very structures involved in the impedance control (OHCs and Deiters' cells). When the intensity increases, the ensuing impedance regulation will reduce the BM-to-hotspot pickup at any frequency: not only for near-CF tones that excite local IHCs and OHCs well, but also for off-CF tones that don't. The low-frequency compression is thus demoted to a side effect that nevertheless exposes an underlying compressive mechanism.

In a similar vein, parametric impedance adjustment may account for the other peculiarity of the hotspot's nonlinearity, its non-monotonic I/O functions for near-CF components (Fig. 6f).

Traveling waves at and beyond CF have already been subject to compression by the OHCs at more basal locations. The local OHCs serve to compress the slightly lower-frequency waves that are on their way to stimulate IHCs at a slightly more apical place. But the mechanism by which the local OHCs exert this control—the parametric impedance adjustment—cannot help also shaping their response to the already compressed CF waves, namely, by adding even more compression. This naturally leads to the hyper-compression observed in the hotspot (but not on the BM). Again, the hotspot motion itself may bear little resemblance to local IHC input, but exposes the mechanism of dynamic range compression in the cochlea, which is one of its two major functions.

Cochlear frequency selectivity and dynamic range compression are two of the cornerstones of auditory processing, and our current understanding of these processes is limited. New measurements, including those presented here, offer a fascinating glimpse of the actual structures performing these tasks, and we are confident that their systematic study will soon lead to a sound understanding of cochlear function.

## Methods

**Animal preparation**. Experiments were performed in accordance with the guidelines of the Animal Care and Use Committee at Erasmus MC, who approved all protocols. Sound-evoked vibrations were recorded from the cochlear partition and ossicles of healthy young female gerbils ($n = 51$, aged 44–188 days, weight range 54–82 g; the main reason for only studying females is that they are more sociable than males, and therefore more convenient to accommodate within Erasmus MC).

Animals were anesthetized using intraperitoneal injections of ketamine (80 mg/kg) and xylazine (12 mg/kg), with no recovery allowed at the end of the experiments. Maintenance (1/4) doses of the anesthetic were given at intervals of between 10 and 60 min, as required to abolish pedal withdrawal reflexes. Animals were tracheotomized, but self-ventilating. Core temperatures were maintained at 38 °C using a thermostatically controlled heating pad. The pinna and external meatus of the left ear was retracted and a $4 \times 6$ mm$^2$ wide opening was made into the postero-lateral bulla to expose the basal aspects of the cochlea, including the round window and the middle-ear ossicles (Fig. 1a). Additional heating of the environment around the animals head was provided using a thermostatically controlled infrared lamp, such that the temperature at the edge of the open bulla was maintained at 34–35 °C. A paper wick was used to prevent any buildup of exudate in the round window recess; the wick was positioned so as not to impede movements of the round window membrane itself (see Fig. 1a).

All experiments were performed in a sound-proof chamber, with the animals supported in a goniometric cradle mounted on a vibration isolated table. Imaging and vibration measurements were made in the first turn and hook region of the cochlea, as viewed through the intact round window membrane, and from the footplate and/or posterior crux of the stapes in the middle ear. All measurements were made under open-bulla conditions, but the cochlea itself was intact. The physiological condition of the cochlea was assayed using compound action potential measurements from a silver wire electrode placed on the wall of the basal turn of the cochlea[65].

**OCT imaging and vibrometry**. A spectral domain OCT system (Thorlabs Telesto TEL320C1, operating with a central wavelength of 1300 nm) was used for interferometric imaging and vibration measurements. The system provided cross-sectional (B-scan) and axial images (A-scans and M-scans) that were triggered externally using TTL pulses phase-locked to an acoustic stimulation system (Tucker Davies Technologies system III). The sampling rate of both the sound-generation and OCT-recording system was 111.6 kHz. The M-scans consist of time-stamped optical spectra whose Fourier transforms provide depth-resolved images (i.e., maps of scattering intensity versus axial depth) and vibration information (i.e., maps of scatterer-displacement versus axial depth). Each optical spectrum (A-scan) had 2048 samples covering the ~210 nm bandwidth of the interferometer's light source; each instantaneous axial image and vibration map correspondingly had 1024 spatial "pixels" covering the instrument's ~3.5 mm depth-of-field (i.e., $z$-range). The optical recording system had an axial point spread function with a FWHM of ~6 μm, a lateral resolution (in the $xy$ plane) of 13 μm, and a linear operating range of >500 μm (all assessed in air, with a refractive index of 1). For comparison with other studies, e.g., refs.[22,23,25–28], the theoretical axial resolution of our OCT system was ~2.9 μm in perilymph (with an assumed refractive index of 1.3); the numerical aperture of the imaging lens was 0.055, and the amount of light incident on the cochlea was ~3.7 mW. The sensitivity of the A-scan's phase-spectra to vibration permitted measurement noise-floors that ranged from ~30 pm/√Hz in the cochlea down to ~3 pm/√Hz in the middle ear.

Intra-cochlear images (B-scans) were formed by scanning the OCT across a series of parallel, quasi-radial (but typically far from transverse) planes that sectioned the cochlear partition at different longitudinal positions (cf. Fig. 1b; in this example, quasi-radial B-scans were made at each of the positions indicated by the red dots). The spacing between the initial measurement planes was ~50–100 µm. Readily identifiable loci within each of the resultant images (e.g., the characteristic junction between the arcuate and pectinate zones of the BM; cf. Fig. 1c, d) were then used to define the true longitudinal, radial and transverse axes of the partition post-hoc. Images were subsequently compensated for the oblique angles of incidence of the recording beam, and for the refractive index of the intra-cochlear fluids, which was assumed to be 1.3.

Vibration measurements (M-scans) were made by aiming the OCT at one particular $x,y$ locus and recording a contiguous series of ~1.5 million A-scans, before moving on to repeat the same procedure at subsequent sites if and when required (cf. Fig. 1b). For radial mapping purposes (cf. Figs. 1e, 2–4, and Supplementary Figs. 2, 3), measurement positions were typically spaced at intervals of between 6 and 12 µm in the $xy$ plane (e.g., see blue dots in Fig. 1b).

It is important to note that the alignment of the measurement beams in most of our OCT recordings was far from perpendicular to any of the cochlea's principal anatomical axes (cf. Figs. 1b, 8): the measurements that we made from each angle should therefore be sensitive to structural movements in all three cochlear dimensions (radial, transverse, and longitudinal). Some fundamental consequences of this three-dimensional sensitivity are explored further in the Supplementary Note 4 (cf. Supplementary Fig. 4).

**Post-hoc skew correction**. Vibration measurements were corrected for the refractive index of the fluid, but not (directly) for the angle of incidence. Post-hoc corrections were made for the longitudinal skew of the measurement sites with sectioning depth (i.e., $z$ in Fig. 1b): both the amplitude and the phase of the recorded vibrations were adjusted according to preparation-specific tonotopic maps derived by interpolation from the previous (longitudinal) series of measurements at well identified anatomical loci (see above, and red dots in Fig. 1b). The skew corrections result in maps of vibration that correspond to planes that are perpendicular to the longitudinal axis of the organ of Corti, even though the measurement beam was not perpendicular to this axis. Without this correction, the slanted orientation of the cross sections would introduce a parallax, i.e., a confounding between transverse and longitudinal dimensions. The deeper structures in most of our cross-section are located further in the apical direction (Fig. 1b), resulting in larger phase lags and lower CFs than the more superficial structures. This parallax is undone by the correction.

**Acoustic stimulation**. To optimize the amount of useful information that could be collected in each experiment, multi-tone "zwuis" complexes[66] were used as acoustic stimuli. Each complex had between 30 and 50 spectral components spanning most of the animal's hearing range (typically 0.2 to ~30 or ~50 kHz). All components had approximately equal amplitudes, but phases that were randomized across frequency. Each stimulus was presented for 12 s, and inter-stimulus intervals were ~1 min. The stimuli were coupled into the exposed ear-canal using a pre-calibrated, closed field sound-system. Sound pressure levels are expressed in decibels re: 20 µPa (i.e., dB SPL) per spectral component.

**Response analysis**. Responses were analyzed by Fourier transformation of the vibration waveforms derived from contiguous groups of 3 pixels in each M-scan, where each pixel covers a depth of ~2.7 µm in the fluid-filled spaces of the cochlea, and ~3.5 µm in the air-filled spaces of the middle-ear. Responses from intra-cochlear locations were corrected for path-length differences originating in the movements of the overlying round window membrane (i.e., the air–fluid interface in our preparations; cf. ref.[67]). The statistical significance of each response component was assessed using Rayleigh tests of the component's phase stability across time[66]. Two-dimensional interpolation was used for spatial mapping purposes (see Supplementary Fig. 1).

**Data availability**. The data presented in Figs. 1–8 of this study can be found at https://doi.org/10.6084/m9.figshare.c.4147139.

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

## Acknowledgements

We thank Drs. Gerard Borst, Mary Palmer, and Natalie Cappaert. This work was supported by the Netherlands Organization for Scientific Research, ALW 823.02.018, and an EU Horizon 2020 Marie Sklodowska-Curie Action Innovative Training network, H2020-MSCA-ITN-2016 [LISTEN - 722098].

## Author contributions

All authors contributed extensively to this work. N.C. and M.v.d.H. devised and adapted the experimental techniques and wrote the paper. N.C. performed most of the experiments. A.V. collated and analyzed data, as well as performed experiments. M.v.d.H. obtained funding, ran the lab, developed software for all aspects of the study, and analyzed all of the data.

## Additional information

**Competing interests:** The authors declare no competing interests.

