## [Peer Review File · Nature Communications]

Reviewers' comments:

Reviewer #1 (Remarks to the Author):

This paper presents vibration measurements obtained with very high spatial resolution of structures (basilar membrane and cells in the organ of Corti) near the extreme basal end of intact gerbil cochleae. The data are exquisite and extremely valuable, deserving prompt publication. The authors focus on the largest vibrations, located in a "hotspot" in the organ of Corti, with properties that differ substantially from those of nearby basilar-membran sites. Those results are tantalizingly similar to those described by Ren et al. (PNAS 2016) for "reticular lamina" sites at the base of the mouse cochlea, which also differ from basilar-membrane vibrations. In both preparations (mouse and gerbil), the organ of Corti vibrations are physiologically vulnerable and larger and more broadly tuned than basilar-membrane vibrations. However, responses in the gerbil differ from those in mouse in that they display hypercompressive behavior near the characteristic frequency and that at low frequencies adjacent basilar-membrane and organ of Corti move in phase. One wishes that Cooper et al. would discuss these differences. [In the present text, they already discuss anomalous antiphase behavior in gerbil which depends on the angle of measurements at sites very near the extreme basal end of the cochlea. Conceivably, this finding could be related to the antiphase behavior in mouse.]

Reviewer #2 (Remarks to the Author):

Mammalian cochleae have exquisite sensitivity, sharp tuning, and wide dynamic range for processing and detecting different environmental sounds. The mechanism underlying these remarkable capabilities has been a central topic for auditory scientists. In the current study, the authors used a spectral domain optical coherence tomography system for imaging and vibration measurements of the cochlear partition in intact living gerbil cochleae. By mapping sound-induced vibrations on the structural image, the authors found that the largest vibrations were within a tightly delineated 'hotspot' in the midway between the basilar membrane and the reticular lamina. Hotspot vibrations were most prominent at low sound level; physiologically vulnerable; more broadly tuned than basilar membrane vibrations, and longitudinally oriented at low frequencies. It was concluded that structural coupling between the Deiters cells and outer hair cells in the hotspot region 'funnels' sound-evoked motion longitudinally and that the mechanically sensitive hair cells control this mechanism to optimize cochlear tuning and compression.

The major claim of this paper is that a sound-induced vibration hotspot was found in an unexpected cochlear location and this experimental finding reveals a new micromechanical mechanism, i.e., longitudinal funneling of sound-induced motion in the mammalian cochlea. The main technical advance of this study is the superior spatial resolution of imaging and vibration measurements, which allows the authors to correlate the vibration map with the microstructures of the cochlear partition. The measurements are novel and data will be highly interesting to auditory scientists, biophysicists, and broad readership of Nature Communications. The reported new data and the proposed novel idea will likely influence fundamental thinking in the field of cochlear mechanics.

The manuscript can be improved by addressing following comments.

1. While it is explicitly stated in the abstract as "The largest vibrations were observed within a tightly delineated 'hotspot' in an unexpected location, midway between the basilar membrane and the reticular lamina", the data show that the location of the vibration hotspot varies with the sound level (Fig. 3), cochlear conditions (Fig. 4), and the stimulus frequency (Supplementary Fig. 2). A more accurate and complete description of the observation should be provided.

2. Although the authors make it clear that the vibration hotspot is inconsistent with established active cochlear models, it is difficult to comprehend the following text; "...that the hot spot motion results from structural constraints that require physiological integrity, and the that OHCs act as mechanical regulators (17) rather than as the direct sources of hotspot vibrations (and/ or BM vibrations near CF). Supporting evidence for such parametric control comes from temporal sluggishness cochlear compression (41)". This statement likely depends on authors' previous work cited in references 17 and 41. A brief review of those studies should be helpful in improving the readability of the paper.

3. A key component in the authors' alternative theory is that the parametric impedance control is performed by the OHCs that change their stiffness. However, it has been well documented that outer hair cells can generate receptor potential in response to sound stimulation and the transmembrane potential can result in hair cell length change through prestin-mediated somatic motility. The authors should explain why expected anti-phase motions at both ends of the outer hair cells were not observed in this study.

4. Fig. 8, panel C: It is not obvious how the viewing angle affects phase of measured motion. Should motion measured from viewing angle ϕ_2 have a 0.5-cycle phase difference from that measured from viewing angle ϕ_1 ?

5. Considering its importance for the data presentation, "structural framework of Corti's organ" should be clearly defined.

6. Methods: Why were female gerbils and multi-tone stimulation used in this study.

7. Methods, OCT imaging and vibrometry, line 2: "operating at 1300 nm" should be "operating at a central wavelength of 1300 nm".

8. Supplementary Fig. 4, line 7 in legend: "(E)" should be "(C)".

9. To help the readers reproduce the reported work, the author should provide more technical information, such as the scanning rate, parameters of the external trigger signal and objective lens.

Reviewer #3 (Remarks to the Author):

This is a story of very interesting data buried beneath rigid concept and speculation. There is even a sub section in the results, possibly regarded by the authors as their most important contribution, which is entire speculation. The paper was made very difficult to read through close line spacing and the absence of page and line numbers. It was also necessary for us to read every paper cited by the authors to check the accuracy of claims made by the authors on the outcomes of the papers they cited. Regrettably, this was not always the case. The authors also overlooked papers that had reached similar conclusions, although based on different evidence. There were also claims about the increased resolution of the methods that appear not to be true, even when comparing data obtained nearly two years ago. Presentation of the data was also not satisfactory, and there is a need for standardization. To us it was good data spoiled.

Specific comments.

Abstract

map sound-evoked vibrations on to the anatomical structure of the hearing organ with unprecedented spatial resolution

Is this true? Images presented in Fig1C and 2A appear just as difficult to discern as for example Lee...Oghalai 2016, Fig 1.

Omit 'unprecedented spatial resolution' because you can't justify this.

According to your methods: 'The recording system had an axial point spread function with a FWHM of $\sim 6 \mu\text{m}$, a lateral resolution (in the xy plane) of $\sim 10 \mu\text{m}$, and a linear operating range of $> 500 \mu\text{m}$ (all assessed in air, with a refractive index of 1).

According to Lee et al's methods, their spatial resolution is similar to yours. 'we use the FWHM diameter of $10.9 \mu\text{m}$ as a better indication of imaging resolution. Therefore, in water, which has a refractive index of ~ 1.3 at 1300 nm , the theoretical lateral imaging resolution is $8.4 \mu\text{m}$. We have measured the lateral resolution of our VOCTV system and found it to be reasonably close to this at $9.8 \mu\text{m}$ (Lee et al., 2015). With such a low NA objective, the axial resolution is set by the bandwidth of the laser source, and we measured this to be $11.4 \mu\text{m}$ in water (Lee et al., 2015).'

These comments also pertain to the beginning of your results section. Not able to specify page or line number.

...The largest vibrations were observed within a tightly delineated 'hotspot' in an unexpected location,

Why unexpected?

The location coincides with that of the OHCs, DCs, and IPCs. These are the expected locations of the cochlear amplifier and its structural framework.

Omit 'unexpected'

.... and longitudinally oriented at low frequencies.

It is understood from the methods that measurements are made in the axis of the measuring beam. Thus, it is not clear from the statement if the vibrations are confined to the longitudinal mode, that the vector sum is in (dominated by) the longitudinal mode, or that there are radial and transverse modes, which can be dominant, that are associated with a longitudinal mode. Perhaps make this section clearer?

via a novel, longitudinal vibration mode, and that the mechanically-sensitive hair cells control this funnel to optimize cochlear tuning and compression. This idea is not novel. E.g. see Geisler and Sang, 1995; Russell, Nilsen, 1997; Yoon et al., 2011, who also suggested longitudinal coupling via the DC phalangeal processes and OHCs. Omit 'novel'.

Main Text

Page 1. "...but evidence of power amplification in real cochleae is sparse and indirect (13–15)".

What is wrong with Lukashkin et al. (2007) which is not cited, and which shows negative damping of the BM response at the CF? Negative damping is clear evidence of energy production. Also see Gummer et al. (2017), Jenkins (2013).

Vibration hotspots

Fig. 3. Because the reference value for the dB scale and location of the reference both change between panels it is difficult to appreciate how vibrations change at a particular point. Please use a single reference value for dB scale and a single location.

Broader tuning of the hotspot

Fig. 5. The same frequency range should be used for the axes in A and B.

Figs. 5 and 6. There is a problem with data presentation here. The authors plot responses for a single location within their hotspot on both figures only to say later (Fig. 7) that responses within the hotspot change dramatically. What is the point of illustrating responses for a single location if responses for other locations are very different? Please consider an effective and more convincing way of presenting data.

One might expect frequency selectivity in the hotspot to match neural tuning at least as well as the BM does.

And

BM vibrations is very similar to tuning in IHCs and the auditory nerve

These statements are misleading and the second is wrong. You are presumably measuring transverse motion of the BM. When Naryan et al., (1998) measured the same parameter, they find a mismatch of about 30 dB SPL in the low frequency tail when they compared neural and BM displacement frequency tuning curves. This the same as that reported by Russell et al., (1995) when comparing BM displacement and IHC receptor potential tuning curves in the 18 kHz region of the guinea pig cochlea. Tuning curves obtained from the OHCs were almost identical to those of the BM. Misquoting Naryan's findings won't provide an explanation for your findings. The vital measurement missing from your data is the shear displacement imparted to the OHCs that is directly translated into transverse motion of the OHCs. Measures of this have already been published (e.g. Kössl and Russell, 1992).

Although a direct comparison with auditory nerve data from this region of the gerbil cochlea is difficult, the BM's tuning is the closest to matching typical high frequency neural tuning curves, whose tip/tail ratios are >40 dB (26,27).

Again, this is not correct, and an inadequate basis for comparison.

The tips of BM displacement, OHC and IHC voltage, and neural threshold tuning curves measured under similar conditions, matching criteria, from the same location on the BM are closely similar (Naryan et al., 1998; Russell et al., 1995). At a point approximately half octave below the CF, the tails of the tuning curves behave in different ways. The tails of the BM and OHC tuning curves are closely similar and asymptote at a level about 40 dB SPL above the tip. Neural and IHC tuning curves are also similar, they never quite asymptote but are > 60 dB SPL above the tip. It is thus true to say that the tips of tuning curves based on direct BM mechanical and electrical measurements from the hair cells and nerve fibres are closely similar, but the tails of the tuning curves differ considerably. This point should be clarified in the paper. See also M.A. Cheatham, P. Dallos *Hearing Research* 108,(1997),191-212 for an excellent treatment of this point.

Hyper-compression in the hotspot

Similar, findings, what you call 'hypercompression' have already been reported for the level functions of OHC receptor potentials (e.g. Kossl and Russel 1992). It is likely that OHC voltage dependent motility will be determined by transmembrane potentials across the basolateral membranes of the OHCs (e.g reviewed by Ashmore 2008). In the guinea pig cochlea, strong compression of ac receptor potentials is associated with the appearance of dc receptor potentials in the OHCs. One suspects that tonic displacements might also be recorded in mechanical measurements in the vicinity of the OHCs and DCs. Is this the case?

Evidence for longitudinal hotspot motion

Longitudinal movement of the OHC basal pole was observed by Karavitaki and Mountain (2007) due to simple rotation of excited OHCs around their apical pole. It was not "longitudinal and radial motion of OHC soma" as cited by the authors on page 12. The origin of this rotational movement is obvious. RL, OHC and phalangeal process of the Deiters' cell contacting the OHC form a triangle with two edges of the triangle, which correspond to the RL and phalangeal process, being very stiff. As a result of this arrangement, shortening of the OHC causes it to rotate around the RL attachment point towards the cochlea base. No longitudinal movement of the RL, similar to that shown in the supplementary movie presented by the authors, was observed by Karavitaki and Mountain. Indeed, it is doubtful if this movement can exist. This is because if neighbouring regions of the RL, which is very stiff, should make longitudinal movement, then they will have to make them at different phases due to phase differences along the traveling wave. This is obviously impossible due to the high stiffness of the RL, which make all further discussions about excitation of OHCs due to longitudinal movement of the hotspot/RL fallacious.

This section is entire speculation and should not appear in the results section.

Page 12. "All four of the above predictions, derived from the hypothetical contribution of longitudinal motion, are confirmed in the spatial profiles of Fig. 7".

The authors fail to mention that their predictions are also observed for purely transversal movement of the RL (Fig. 3 in Ren et al. (2016)). Surprisingly, Ren et al. is cited in the manuscript but for different reasons. Phase lead of the RL at low frequencies is more than 0.25-cycle in Ren et al. but it may well be a consequence of recording in the high-frequency region of mouse cochlea.

Discussion

Based on the hotspot's larger motion and closer vicinity to the IHCs, it is tempting to assign hotspot vibrations a greater functional relevance (than BM vibrations) to hearing. The hotspot's poorer frequency selectivity (Fig. 5) and non-monotonic IO functions (Fig. 6)(25,26), contradict this: auditory nerve tuning is at least as sharp as BM tuning. This is a superficial and inaccurate account of a complex subject. As noted above the tips of cochlear mechanical, electrical, and neural tuning curves are similar but the tails differ. Please see pages 207-218 of M.A. Cheatham, P. Dallos Hearing Research 108,(1997),191-212 for an in depth discussion of this point, and modify this section of the text accordingly

non-monotonic (35,36). IO functions are rarely seen in IHC or neural recordings

This is not true. See Fig 5 of Dallos, 1985, and numerous figures in the Russell and Sellick, Cody and Russell, Kossl and Russell papers. Modify or omit the text accordingly.

In stark contrast, longitudinal motion may well be sensed by OHC stereocilia: OHCs have characteristically V or W shaped stereociliary bundles, each wing of which is oriented at a considerable angle to the cochlear spiral (28,33) and may therefore act independently to sense (and rectify) an excitatory stimulus (e.g. 8).

These reverse polarity channels do not survive in the adult cochlea and would not signal a receptor current (see 8). Thus, a longitudinally conducted displacement of the reticular lamina, or other elements of the cochlear partition would increase the open probability of mechano sensitive channels on one wing of the OHC bundle and decrease the open probability on the other wing, depending on the phase of the periodic movement. So we can't see this working.

Page 13. "The very existence of a frequency-, intensity- and physiologically-dependent motion hotspot within the organ of Corti suggests that the classical way of modelling cochlear mechanics, with the BM as the dominant elastic structure and the surrounding fluid as amorphous mass load (31,38) is incomplete."

This is one of several 'Straw Man' arguments found in the manuscript. There is a good understanding within the field that movement within the organ of Corti is complex and frequency-, intensity- and physiologically dependent. There are quite a few models which include movements within the organ of Corti, it is just the experimental data that are sparse. Reduction of cochlear models to "the BM as the dominant elastic structure and the surrounding fluid" is generally made when this level of reduction is legitimate to answer a particular question and limitations of this approach are well understood.

Page 13. "The wideband character of the hotspot's motion (and its nonlinearity) contrasts with the fundamental feature of active models that "the region of activity is spatially limited" (13)."

This is another "Straw Man" fallacy. Reference 13 is 18 years old. There is a good understanding nowadays that OHC amplify vibrations of the BM within a limited frequency range simply because timing of the OHC forces is optimal within this frequency range. There is, however, no restriction on OHC excitation only within this limited frequency range. OHCs can well be excited over much broader frequency range but they are not able to undamp BM responses over this much broader range. Cited Ren et al. (2016) is a good illustration of this

concept.

Page 13. "BM motion is smaller than hotspot motion, and the 0.25-cycle BM/hotspot phase difference at low frequencies is inconsistent with the high speed of OHC length changes." But the OHCs are excited by relative displacement between the TM and RL and any phase relationship could happen. Also see our comment above.

Page 14. Paragraph starting "We propose that hotspot motion plays an important functional role in enhancing cochlear frequency selectivity."

Provides quite vague description of what the authors think is happening during travelling wave propagation. What do you mean by "to become more focused than it is in the dead organ"?

Page 14. "On the other hand, reports on distortion product emissions have provided indirect evidence of an origin of distortion products remote from the CF region, at a more basal site (45)."

These results are readily explained by a shallow high-frequency slope of suppression tuning curves (e.g. see Nam and Guinan (2018)).

Page 14. Paragraph starting "The wideband compression observed in the hotspot is puzzling from a functional point of view."

and the next paragraph. It is not puzzling at all if you separate the concepts of amplification and OHC excitation as indicated in our comment above. The authors have to accept later in the same paragraph that "BM motion represents the "true" carrier of the low-frequency wave" but they are still too shy to accept that amplification of the BM responses and OHC excitation and resultant compression of responses within the organ of Corti are different phenomena.

Last three paragraphs of the Discussion. The idea of parametric control of the OHC operation/impedance matching is not novel and the authors should provide references to relevant publications.

Authors responses (**bold**) to reviewers' comments (*italic* / **bold italic**):

Reviewer #1 (Remarks to the Author):

*This paper presents vibration measurements obtained with very high spatial resolution of structures (basilar membrane and cells in the organ of Corti) near the extreme basal end of intact gerbil cochleae. **The data are exquisite and extremely valuable**, deserving prompt publication. The authors focus on the largest vibrations, located in a “hotspot” in the organ of Corti, with properties that differ substantially from those of nearby basilar-membran sites. Those results are tantalizingly similar to those described by Ren et al. (PNAS 2016) for “reticular lamina” sites at the base of the mouse cochlea, which also differ from basilar-membrane vibrations. In both preparations (mouse and gerbil), the organ of Corti vibrations are physiologically vulnerable and larger and more broadly tuned than basilar-membrane vibrations. **However, responses in the gerbil differ from those in mouse in that they display hypercompressive behavior near the characteristic frequency and that at low frequencies adjacent basilar-membrane and organ of Corti move in phase. One wishes that Cooper et al. would discuss these differences.** [In the present text, they already discuss anomalous antiphase behavior in gerbil which depends on the angle of measurements at sites very near the extreme basal end of the cochlea. Conceivably, this finding could be related to the antiphase behavior in mouse.]*

We thank reviewer #1 for this praise, and have extended our comparisons with the studies of Ren et al. (and others) in the revised MS (see second paragraph of Discussion). We have also re-organised the final section of our Results (including Figures 7,8) and related Discussion to place a greater emphasis on the phase relationship between the BM and OC motion, in order to clarify a point that this reviewer may have missed – the BM and OoC (either ‘hotspot’ or RL) do NOT move “in phase” at low frequencies (at least when the cochlea is alive).

Reviewer #2 (Remarks to the Author):

Mammalian cochleae have exquisite sensitivity, sharp tuning, and wide dynamic range for processing and detecting different environmental sounds. The mechanism underlying these remarkable capabilities has been a central topic for auditory scientists. In the current study, the authors used a spectral domain optical coherence tomography system for imaging and vibration measurements of the cochlear partition in intact living gerbil cochleae. By mapping sound-induced vibrations on the structural image, the authors found that the largest vibrations were within a tightly delineated 'hotspot' in the midway between the basilar membrane and the reticular lamina. Hotspot vibrations were most prominent at low sound level; physiologically vulnerable; more broadly tuned than basilar membrane vibrations, and longitudinally oriented at low frequencies. It was concluded that structural coupling between the Deiters cells and outer hair cells in the hotspot region 'funnels' sound-evoked motion longitudinally and that the mechanically sensitive hair cells control this mechanism to optimize cochlear tuning and compression.

*The major claim of this paper is that a sound-induced vibration hotspot was found in an unexpected cochlear location and this experimental finding reveals a new micromechanical mechanism, i.e., longitudinal funneling of sound-induced motion in the mammalian cochlea. The main technical advance of this study is the superior spatial resolution of imaging and vibration measurements, which allows the authors to correlate the vibration map with the microstructures of the cochlear partition. **The measurements are novel and data will be highly interesting to auditory scientists,***

biophysicists, and broad readership of Nature Communications. The reported new data and the proposed novel idea will likely influence fundamental thinking in the field of cochlear mechanics.

We thank reviewer #2 for this opinion.

The manuscript can be improved by addressing following comments.

*1. While it is explicitly stated in the abstract as "The largest vibrations were observed within a tightly delineated 'hotspot' in an unexpected location, midway between the basilar membrane and the reticular lamina", the data show that **the location of the vibration hotspot varies with the sound level (Fig. 3), cochlear conditions (Fig. 4), and the stimulus frequency (Supplementary Fig. 2). A more accurate and complete description of the observation should be provided.***

We have revised the MS as requested, by adding two new sentences to the Results section describing Figs. 3, 4 and Supp. Fig 2. We have tried to avoid giving the impression which seems to have been picked up (by reviewer #2) above: namely that the hotspot is a movable feast. We have emphasized from the outset that the focus of the current paper is the hotspot that occurs in the OHC/Deiters' cell area.

*2. Although the authors make it clear that the vibration hotspot is inconsistent with established active cochlear models, **it is difficult to comprehend the following text; "...that the hot spot motion results from structural constraints that require physiological integrity, and the that OHCs act as mechanical regulators (17) rather than as the direct sources of hotspot vibrations (and/ or BM vibrations near CF). Supporting evidence for such parametric control comes from temporal sluggishness cochlear compression (41)". This statement likely depends on authors' previous work cited in references 17 and 41. A brief review of those studies should be helpful in improving the readability of the paper.***

Due to space concerns, and reviewer #3's concerns that these ideas "are not novel", we have revised the MS only slightly here. We have added one much older reference, and a few key words as descriptive pointers to explain that: "...OHCs act (on a slower timescale) as mechanical regulators (^{58,21}), rather than as rapid, direct sources of hotspot vibrations..."

*3. A key component in the authors' alternative theory is that the parametric impedance control is performed by the OHCs that change their stiffness. However, it has been well documented that outer hair cells can generate receptor potential in response to sound stimulation and the transmembrane potential can result in hair cell length change through prestin-mediated somatic motility. **The authors should explain why expected anti-phase motions at both ends of the outer hair cells were not observed in this study.***

We have revised the modeling section of our Discussion to clarify two potential explanations for this apparent failure: (1) such anti-phasic motion does not occur in real, acoustically stimulated cochleae (i.e. the so-called cochlear amplifier doesn't work in this way) and (2) it does occur, but it is dwarfed by other motions (such as those with the near 'quadrature' phase-shifts that we do observe).

4. Fig. 8, panel C: It is not obvious how the viewing angle affects phase of measured motion. Should motion measured from viewing angle ϕ_2 have a 0.5-cycle phase difference from that measured from viewing angle ϕ_1 ?

Yes, at least in cases where the longitudinal (horizontal) component of the motion dominates the overall motion. We have redistributed the original material of Fig. 8 and include a new plot in the revised Fig. 8C to graphically illustrate the phase-relationships for a situation that we believe may be relevant for our own low-frequency mechanical measurements. The overall situation is more complex than the relationship between viewing angle and anatomy might suggest, however, as the (unknown) underlying motion is 'free' to project onto the anatomy independently of the viewing angle. This double-projection permits a wide variety of potential relationships between actual motion and the recorded motion. In order to simplify the MS and avoid obfuscating our paper's main message(s), we have relegated much of this discussion to the Supplementary Information.

5. Considering its importance for the data presentation, "structural framework of Corti's organ" should be clearly defined.

We agree, and have inserted our own definition into the revised MS (see introductory paragraph of the Results section).

6. Methods: Why were female gerbils and multi-tone stimulation used in this study.

We have revised the methods section of the MS to clarify these points: There is no 'scientific' rationale to study only females, but from an animal welfare perspective, female gerbils are more sociable, and hence more convenient to rear and house than males. Multi-tone stimuli permit rapid characterization of response properties across a broad frequency range.

7. Methods, OCT imaging and vibrometry, line 2: "operating at 1300 nm" should be "operating at a central wavelength of 1300 nm".

Agreed - MS revised as requested.

8. Supplementary Fig. 4, line 7 in legend: "(E)" should be "(C)".

Agreed – thank you. Due to the more major revisions requested by Reviewer #3, however, Supplementary Fig. 4 has now been deleted/replaced, with parts of the old Figure (panels A-C) now incorporated into Fig. 8 of the main MS.

9. To help the readers reproduce the reported work, the author should provide more technical information, such as the scanning rate, parameters of the external trigger signal and objective lens.

Agreed - MS revised as requested (see various additions to Methods: OCT imaging and vibrometry). One significant omission that we noted on re-reading our Methods section was a description of the corrections that we have applied to compensate for the sound-evoked movements of the round window membrane, which can (and do) affect the apparent motion of structures lying deeper into the cochlea. The procedure that we used to do this has been refined considerably since the first draft of our MS, and is described in detail in the revised Methods section. Applying the new corrections has not changed any of our findings, but it has resulted in very minor changes in the data that we present graphically. Typically, these changes amount to less than 0.1 dB.

Reviewer #3 (Remarks to the Author):

*This is a story of very interesting data buried beneath rigid concept and speculation. There is even a sub section in the results, possibly regarded by the authors as their most important contribution, which is entire speculation. **The paper was made very difficult to read through close line spacing and the absence of page and line numbers.** It was also necessary for us to read every paper cited by the authors to check the accuracy of claims made by the authors on the outcomes of the papers they cited. Regrettably, **this was not always the case.** The authors also **overlooked papers** that had reached similar conclusions, although based on different evidence. There were also **claims about the increased resolution of the methods that appear not to be true**, even when comparing data obtained nearly two years ago. **Presentation of the data was also not satisfactory**, and there is a need for standardization. **To us it was good data spoiled.***

We disagree with the majority of these comments, but we are pleased that the review acknowledges some ‘good’ and/or ‘very interesting’ data. It is very likely that we will have overlooked some papers that may have reached similar conclusions, but almost by definition we are not aware of any, and reviewer #3 mentions only one that we were unaware of (Jenkins, 2013, which we are still unable to track down). The implication that we misquote the literature is disturbing, but (in our view) unsubstantiated. We believe that our MS cites a representative and unbiased selection of high-quality literature, without unfairly twisting the meaning of anyone’s findings (although we can see how we have misled reviewer #3 in the instance of reference #9). To the best of our knowledge, our claims about resolution are also true, as addressed below.

We have increased the line spacing and added page and line numbers to the revised MS, as required/requested.

Specific comments.

Abstract

map sound-evoked vibrations on to the anatomical structure of the hearing organ with unprecedented spatial resolution

***Is this true?** Images presented in Fig1C and 2A appear just as difficult to discern as for example Lee...Oghalai 2016, Fig 1.*

Omit ‘unprecedented spatial resolution’ because you can’t justify this.

According to your methods: ‘The recording system had an axial point spread function with a FWHM of $\sim 6 \mu\text{m}$, a lateral resolution (in the xy plane) of $\sim 10 \mu\text{m}$, and a linear operating range of $>500 \mu\text{m}$ (all assessed in air, with a refractive index of 1).

According to Lee et al’s methods, their spatial resolution is similar to yours. ‘we use the FWHM diameter of $10.9 \mu\text{m}$ as a better indication of imaging resolution. Therefore, in water, which has a refractive index of ~ 1.3 at 1300 nm , the theoretical lateral imaging resolution is $8.4 \mu\text{m}$. We have measured the lateral resolution of our VOCTV system and found it to be reasonably close to this at $9.8 \mu\text{m}$ (Lee et al., 2015). With such a low NA objective, the axial resolution is set by the bandwidth of the laser source, and we measured this to be $11.4 \mu\text{m}$ in water (Lee et al., 2015).’

We believe that our claim is true, but we would be happy to omit it from the abstract if necessary (we have not done so yet). The (theoretical) spatial resolutions of Lee et al.’s technique and our own are similar in the lateral dimensions, but in the third dimension (axially) ‘ours’ is almost twice as good as ‘theirs’. As noted in our MS, as well as by Lee et al., the imaging quality of any system depends not only on such theory, but also on practice: that is, how and where the measurements

are actually made. So while the bandwidth of our light source is twice that used by Lee et al (giving us the theoretical 2x better axial resolution), the vast majority of the improvement that we see over Lee et al. actually comes from approaching the partition through the round window membrane, rather than through the bony shell of the cochlea. We are aware of other studies that achieve even greater resolution than our own, but to date (as far as we are aware) these have not been applied to the intact, living cochlea. The improvement over previously published studies in the intact, living cochlea, which is the basis of the very specific claim of made in our report (page 5, lines 68-69 of the revised MS), is fully acknowledged by John Oghalai, the senior and corresponding author of the Lee et al. studies that reviewer #3 focusses on. We have amended the text of the revised MS (in the Methods and Discussion sections) to clarify and support our (very specific) claim of unprecedented resolution.

These comments also pertain to the beginning of your results section. Not able to specify page or line number.

...The largest vibrations were observed within a tightly delineated 'hotspot' in an unexpected location,

Why unexpected?

The location coincides with that of the OHCs, DCs, and IPCs. These are the expected locations of the cochlear amplifier and its structural framework.

Omit 'unexpected'

In the interests of brevity, we have amended the abstract as requested. As hinted in our original Discussion, we believe that our own 'expectations' of which structures would move most in the organ of Corti have been severely limited by many of the concepts put forward over the past 30+ years of theoretical research into the nature of the so-called 'cochlear amplifier' – a hypothetical amplifier that was invented to act on the basilar membrane itself. As far as we can see, none of this research included the idea that the bases of the hair cells either would or could move more than both the reticular lamina and the basilar membrane (even though there were clear indications of this in experiments published as early as 2007). We have tried to make the distinction between what conventional models 'predicted' and what our data show much clearer in the revised Results and Discussion sections.

.... and longitudinally oriented at low frequencies.

It is understood from the methods that measurements are made in the axis of the measuring beam.

Thus, it is not clear from the statement if the vibrations are confined to the longitudinal mode, that the vector sum is in (dominated by) the longitudinal mode, or that there are radial and transverse modes, which can be dominant, that are associated with a longitudinal mode. Perhaps make this section clearer?

For simplicity, we have removed this statement from the amended abstract. The concept of longitudinal motion is now mentioned only in terms 'strong evidence' and the novel proposal in the final two sentences of the abstract (see below), and this is expanded substantially in the Results, Discussion and Methods sections.

via a novel, longitudinal vibration mode, and that the mechanically-sensitive hair cells control this funnel to optimize cochlear tuning and compression.

This idea is not novel. E.g. see Geisler and Sang, 1995; Russell, Nilsen, 1997; Yoon et al., 2011, who also suggested longitudinal coupling via the DC phalangeal processes and OHCs. Omit 'novel'.

We disagree strongly with the reviewer's comments here – as far as we can see, our suggestion is novel, and the reviewer is simply missing the distinction (that we should perhaps have drawn explicitly in our original MS) between 'longitudinal coupling' and 'longitudinal motion'. Longitudinal coupling is at the heart of many previous modelling studies, including both of those cited by reviewer #3, and is likely to play an important role in the real cochlea (as suggested in all 3 of the studies reviewer #3 cites). To the best of our knowledge, on the other hand, longitudinal motion of structures in the cochlea has only been observed and reported twice previously (references 1 and 34 as cited in the original MS), and is one of the most 'unexpected' findings in our own study: reviewer #3's later comment that longitudinal motion of the RL (specifically) 'is obviously impossible' may provide some insight into why we were not expecting to find such motion (but simultaneously contradicts the direct observations made in references 1 and 34). We believe that our study is the first to demonstrate longitudinal motion at low-to-moderate sound pressure levels in an intact, living cochlea, and/or to suggest that such motion may be central to the normal mechanical function of a healthy cochlea: that is why we consider it to be 'novel'.

We have amended our Discussion to draw an explicit distinction between longitudinal coupling and longitudinal motion.

Main Text

Page 1. "...but evidence of power amplification in real cochleae is sparse and indirect (13–15)". What is wrong with Lukashkin et al. (2007) which is not cited, and which shows negative damping of the BM response at the CF? Negative damping is clear evidence of energy production. Also see Gummer et al. (2017), Jenkins (2013).

We are unsure how to respond to this question. We were very reluctant to cite the Lukashkin et al. paper because it seems likely that there are many things "wrong" with it: a direct rebuttal of its methodology, and the interpretation of its results, has already been published by others (Chan & Nuttall, J. Acoust. Soc. Am. Expr. Lett, 2009). Even if we were to ignore that critique, the paper remains an indirect study, just like the exemplars that we had already cited in the MS. These studies all rely on the fundamental assumption that the (unknown and potentially complex) mechanics of the mammalian cochlea can be represented using lumped parameters and/or point impedances, and assumption which we (and a few other groups) believe is neither warranted nor justified.

We have updated the MS to include references to both Lukashkin et al. (2007) and Chan & Nuttall (2009).

Vibration hotspots

Fig. 3. Because the reference value for the dB scale and location of the reference both change between panels it is difficult to appreciate how vibrations change at a particular point. Please use a single reference value for dB scale and a single location.

The contour plots in Fig. 3 have been rescaled as requested.

Broader tuning of the hotspot

Fig. 5. The same frequency range should be used for the axes in A and B.

The plots shown in Figs. 5A and B illustrate different features of our data, each of which has its own 'natural' relation to frequency: logarithmic scales are used for both amplitude and frequency in panel A, to mimic the cochlea's known near-logarithmic mapping of frequency onto 'place' (and amplitude onto 'loudness'), while linear scales are used for both frequency and phase in panel B to facilitate temporal interpretation of the response. We think that this, combined with aesthetics (the efficient use of journal/Figure space), is a valid reason to leave the limits of the axes as they are.

For reasons explained below, however, the data illustrated in Fig. 5 have been replaced with those from another animal (the same one illustrated more extensively, for different purposes, in Fig. 6).

Figs. 5 and 6. There is a problem with data presentation here. The authors plot responses for a single location within their hotspot on both figures only to say later (Fig. 7) that responses within the hotspot change dramatically. What is the point of illustrating responses for a single location if responses for other locations are very different? Please consider an effective and more convincing way of presenting data.

We are puzzled by this statement / criticism. We cannot find any text that either states or implies that responses within the hotspot change dramatically (and at most frequencies they do not, which we stated explicitly in the original MS). Almost by definition, the responses change dramatically at the boundaries of the hotspot, but as we state (now on page 6, line 100-101) 'the vibration amplitudes change only gradually either inside or outside this boundary'.

The data of Figs. 5 and 6 were also from different animals, so they were not really 'from a single location' in the only other sense that we can imagine the reviewer might mean. To avoid the possibility that other readers might be confused by this type of 'single location' issue, we have replaced the data of Fig. 5 with different data, derived from exactly the same locations in the same cochlea as illustrated in Fig. 6. (We are not completely sure whether this revision is what reviewer #3 is seeking, but we think that it improves the MS anyway).

In relation to Fig. 7, we have inserted text on page 10 (line 194) to try to clarify that "the ~10- μ m-thick transition layer" that we refer to "forms one 'edge' of the hotspot".

One might expect frequency selectivity in the hotspot to match neural tuning at least as well as the BM does.

And

BM vibrations is very similar to tuning in IHCs and the auditory nerve

These statements are misleading and the second is wrong. You are presumably measuring transverse motion of the BM. When Naryan et al., (1998) measured the same parameter, they find a mismatch of about 30 dB SPL in the low frequency tail when they compared neural and BM displacement frequency tuning curves. This the same as that reported by Russell et al., (1995) when comparing BM displacement and IHC receptor potential tuning curves in the 18 kHz region of the guinea pig cochlea. Tuning curves obtained from the OHCs were almost identical to those of the BM.

Misquoting Naryan's findings won't provide an explanation for your findings. The vital measurement missing from your data is the shear displacement imparted to the OHCs that is directly translated into transverse motion of the OHCs. Measures of this have already been published (e.g. Kössl and Russell, 1992).

We disagree with this criticism – our statements were carefully worded, and not intended to be misleading ('very similar' is not meant to imply 'identical'). The points being raised by the reviewer in respect of the cited literature are perfectly valid, but we have not claimed that they are not – and we are certain that we are not “misquoting” Narayan’s very impressive findings - in our MS.

We have adjusted the wording of the two offending statements to try to avoid anyone else being misled as we feel Reviewer #3 has: ‘...match neural...’ has been changed to ‘...approximate IHC, OHC and neural...’, and the sentence that included ‘is very similar to tuning in IHCs’ has been changed to read ‘Mechanical tuning of BM vibrations is remarkably similar (although not entirely identical) to electrophysiological tuning in IHCs, OHCs and the auditory nerve’.

While we agree with Reviewer #3’s point that ‘vital measurements’ are missing from our own data, we disagree that anyone (with the possible exception of Oghalai’s group, e.g. in Lee et al , 2016) has previously *measured* either the shear displacement imparted to the OHCs, nor the resultant (?) transverse motion of the OHCs in a healthy, living cochlea. Many other investigators have discussed these parameters (including Kössl and Russell, 1992), but as far as we are aware no-one has actually *measured* them *in vivo*.

We have added three new citations (alongside Narayan et al., 1998 and Robertson and Patuzzi’s early - 1988 - review) as pointers for background reading on the origins of cochlear tuning: Sellick et al., 1982; Kössl and Russell, 1992; and Robles and Ruggero’s (2001) mechanics review.

Although a direct comparison with auditory nerve data from this region of the gerbil cochlea is difficult, the BM’s tuning is the closest to matching typical high frequency neural tuning curves, whose tip/tail ratios are >40 dB (26,27).

Again, this is not correct, and an inadequate basis for comparison.

Again, we disagree with this criticism. In this case, we have left the text unchanged.

For reasons described above, however, the data of Fig. 5 (and the ‘tip-to-tail’ numbers on which our comparison is based) have been replaced/updated in the text. The changes do not affect the point that we are making.

*The tips of BM displacement, OHC and IHC voltage, and neural threshold tuning curves measured under similar conditions, matching criteria, from the same location on the BM are closely similar (Naryan et al.,1998; Russell et al., 1995). At a point approximately half octave below the CF, the tails of the tuning curves behave in different ways. The tails of the BM and OHC tuning curves are closely similar and asymptote at a level about 40 dB SPL above the tip. Neural and IHC tuning curves are also similar, they never quite asymptote but are > 60 dB SPL above the tip. **It is thus true to say that the tips of tuning curves based on direct BM mechanical and electrical measurements from the hair cells and nerve fibres are closely similar, but the tails of the tuning curves differ considerably. This point should be clarified in the paper.** See also M.A. Cheatham, P. Dallos Hearing Research 108,(1997),191-212 for an excellent treatment of this point.*

We agree with this summary, but feel that it is 100% compatible with what we have already stated in our paper. We have already amended the text to point out that the BM and IHC/AN data are not identical, and added a reference to Kössl and Russell's 1992 study (which compares IHC and OHC tuning in the basal turn of the cochlea), as described above.

Hyper-compression in the hotspot

Similar, findings, what you call 'hypercompression' have already been reported for the level functions of OHC receptor potentials (e.g. Kössl and Russell 1992).

We are aware of the data in Kössl and Russell 1992, but do not agree that they are similar to those in our report. Kössl and Russell report non-monotonic receptor potential IO functions in two tightly defined frequency regions, one slightly below and one slightly above the characteristic frequencies of their cells. In contrast, the IO functions that they report at CF are monotonic. Kössl and Russell also show that the non-monotonicities above and below CF are associated with strong changes in response phase, features which appear quite robust (i.e. insensitive to physiological condition) in their data. This is consistent with the idea that either the hair cells, or (as pointed out by Kössl and Russell) the recordings, pick up different 'drives' at different levels. These characteristics differ in many respects from those that we report for the mechanical hotspot.

We have amended our Discussion to contrast our findings with those of Kössl and Russell (in the OHCs) and Kiang et al. (in the auditory nerve). Specifically, we now state that "strongly non-monotonic IO functions of the type illustrated in Fig. 6 are rarely seen in IHC, OHC or neural recordings (^{46,47}). The few non-monotonic features that have been identified in recordings from the auditory nerve (e.g. ^{48,49}) and OHCs (³⁷) are not restricted to CF, and are rarely as strong as those in the mechanical data of Fig. 6."

It is likely that OHC voltage dependent motility will be determined by transmembrane potentials across the basolateral membranes of the OHCs (e.g reviewed by Ashmore 2008). In the guinea pig cochlea, strong compression of ac receptor potentials is associated with the appearance of dc receptor potentials in the OHCs. One suspects that tonic displacements might also be recorded in mechanical measurements in the vicinity of the OHCs and DCs. Is this the case?

We agree that tonic displacements are likely to occur in this region, but they are very notoriously difficult to isolate, and beyond the scope of the current paper. Related measurements (made using different stimuli) will be published elsewhere.

Evidence for longitudinal hotspot motion

Longitudinal movement of the OHC basal pole was observed by Karavitaki and Mountain (2007) due to simple rotation of excited OHCs around their apical pole. It was not "longitudinal and radial motion of OHC soma" as cited by the authors on page 12. The origin of this rotational movement is obvious. RL, OHC and phalangeal process of the Deiters' cell contacting the OHC form a triangle with two edges of the triangle, which correspond to the RL and phalangeal process, being very stiff. As a result of this arrangement, shortening of the OHC causes it to rotate around the RL attachment point towards the cochlea base. No longitudinal movement of the RL, similar to that shown in the supplementary movie presented by the authors, was observed by Karavitaki and Mountain. Indeed, it is doubtful if this movement can exist. This is because if neighbouring regions of the RL, which is very stiff, should make longitudinal movement, then they will have to make them at different phases

due to phase differences along the traveling wave. This is obviously impossible due to the high stiffness of the RL, which make all further discussions about excitation of OHCs due to longitudinal movement of the hotspot/RL fallacious.

This section is entire speculation and should not appear in the results section.

We disagree strongly with this interpretation. While it is tempting to summarise Karavitaki and Mountains's data as showing "simple rotation" (as Karavitaki and Mountain justifiably did in their own abstract), a closer inspection of their data (e.g. Fig 13 of their paper) and of their text (e.g. 5-line paragraph at top right of their p3309) reveals exactly what reviewer #3 claims "was not observed" by the authors. Related observations, demonstrating both rotation and translation of the OHCs, have also been made under conditions of acoustic (not electrical) stimulation by von Bekesy (see Fig. 12-15 and accompanying text in reference 1). We also note that in the one of the most famous representations of the OHC/RL/DC phalangeal "triangle" referred to by Reviewer #3, where it was first postulated that motile OHCs might induce organ of Corti deformations (Brownell et al., 1985), the primary pivot point for OHC rotations was portrayed as being at the junction with the DC soma, leading to clear longitudinal 'translation' of the RL itself (see Fig. 1c of Brownell et al., 1985). Reviewer #3's claim that longitudinal motion of the RL is precluded (or "is obviously impossible" when the phase differences along the traveling wave are combined with the RL's high stiffness) also seems to overlook von Bekesy's observations. Von Bekesy made these observations almost directly, and reported them as facts (with explicit illustrations) in his book. In a sense, all that we are doing in our paper is making a very similar observation using much a more sensitive technique, lower sound pressure levels, and healthy cochleae.

Page 12. "All four of the above predictions, derived from the hypothetical contribution of longitudinal motion, are confirmed in the spatial profiles of Fig. 7".

The authors fail to mention that their predictions are also observed for purely transversal movement of the RL (Fig. 3 in Ren et al. (2016). Surprisingly, Ren et al. is cited in the manuscript but for different reasons. Phase lead of the RL at low frequencies is more than 0.25-cycle in Ren et al. but it may well be a consequence of recording in the high-frequency region of mouse cochlea.

We do not agree with this statement. Our own measurements are *not* consistent with purely transversal motion, even though our measurements are remarkably (and re-assuringly) similar to and consistent with Ren et al.'s 'two position' data, and include RL-BM phase-differences of greater than 0.25 cycles at low frequencies (which Ren et al. have already shown is *not* unique to the mouse cochlea). Fig. 3 (or any other figure in) the Ren et al. report does not *show* that RL motion is "purely transversal" – Ren et al. simply *assume* transverse motion, and in our view this assumption is questionable.

In our understanding, if the underlying motion were purely transverse, any length changes across the bodies of the OHCs would need to match, or perhaps even exceed, those across the bodies of the Deiters' cells. In our recordings, the differential displacements across the Deiters' cell bodies exceed those across the OHCs.

We have substantially revised the Results and Discussion sections of our MS to try and simplify the presentation of our data and ideas. As also requested by reviewer #2, we now compare our findings more explicitly with those of Ren et al. (and others), and clarify our Discussion of the inconsistencies between our data and classical micromechanical models.

Discussion

Based on the hotspot's larger motion and closer vicinity to the IHCs, it is tempting to assign hotspot vibrations a greater functional relevance (than BM vibrations) to hearing. The hotspot's poorer frequency selectivity (Fig. 5) and non-monotonic IO functions (Fig. 6)(25,26), contradict this: auditory nerve tuning is at least as sharp as BM tuning.

***This is a superficial and inaccurate account of a complex subject.** As noted above the tips of cochlear mechanical, electrical, and neural tuning curves are similar but the tails differ. Please see pages 207-2018 of M.A. Cheatham, P. Dallos *Hearing Research* 108,(1997),191-212 for an in depth discussion of this point, and modify this section of the text accordingly*

We agree that our account is superficial, and that the subject is complex, but we disagree that our account is inaccurate.

As described in our response about "hypercompression" above, we have revised the text to point out that the non-monotonicity observed at CF in our mechanical measurements bears little resemblance to those seen in the OHCs and auditory nerve.

non-monotonic (35,36). IO functions are rarely seen in IHC or neural recordings

This is not true. See Fig 5 of Dallos, 1985, and numerous figures in the Russell and Sellick, Cody and Russell, Kossl and Russell papers. Modify or omit the text accordingly.

We disagree with this criticism, but we have clarified the text in the revised MS ("strongly non-monotonic IO functions, of the type illustrated in Fig 6, are rarely seen in IHC, OHC or neural recordings").

In stark contrast, longitudinal motion may well be sensed by OHC stereocilia: OHCs have characteristically V or W shaped stereociliary bundles, each wing of which is oriented at a considerable angle to the cochlear spiral (28,33) and may therefore act independently to sense (and rectify) an excitatory stimulus (e.g. 8).

*These reverse polarity channels do not survive in the adult cochlea and would not signal a receptor current (see 8). Thus, a longitudinally conducted displacement of the reticular lamina, or other elements of the cochlear partition would increase the open probability of mechano sensitive channels on one wing of the OHC bundle and decrease the open probability on the other wing, depending on the phase of the periodic movement. **So we can't see this working.***

We can see how we have misled reviewer #3 into not believing in the potential mechanism that we are suggesting here, and have clarified the text in the revised MS to avoid this happening to other readers. We were not intending to suggest that the novel MET channels which form the focus of reference 8 in the original MS were responsible for sensing (and rectifying) excitatory stimuli in our preparations. We were suggesting that the regular MET channels in adult OHC's might do this. For such channels (which are not the focus of their paper), the authors of reference 8 clearly state (and show, in their Fig. 1c) that "With fluid-jet orthogonal to the bundle's optimal axis, control MT current has two-harmonic waveform because of alternate stimulation of each wing of the V-shaped bundle". So while reviewer #3 "cannot see this working", the authors of reference 8 clearly can, and have. We are simply combining their observation with our own to propose a (novel?) potential function for the anatomical structure in the living, functional inner-ear. We have revised

the text to specify that it is the control data in Fig. 1C of reference 8 that provides supporting evidence for our hypothesis.

Page 13. "The very existence of a frequency-, intensity- and physiologically-dependent motion hotspot within the organ of Corti suggests that the classical way of modelling cochlear mechanics, with the BM as the dominant elastic structure and the surrounding fluid as amorphous mass load (31,38) is incomplete."

This is one of several 'Straw Man' arguments found in the manuscript. There is a good understanding within the field that movement within the organ of Corti is complex and frequency-, intensity- and physiologically dependent. There are quite a few models which include movements within the organ of Corti, it is just the experimental data that are sparse. Reduction of cochlear models to "the BM as the dominant elastic structure and the surrounding fluid" is generally made when this level of reduction is legitimate to answer a particular question and limitations of this approach are well understood.

We disagree with this criticism, and use the modelling example cited earlier by reviewer #3 ("What is wrong with Lukashkin et al. (2007)") to refute the idea that the limitations of this approach are well understood.

Page 13. "The wideband character of the hotspot's motion (and its nonlinearity) contrasts with the fundamental feature of active models that "the region of activity is spatially limited" (13)."

This is another "Straw Man" fallacy. Reference 13 is 18 years old. There is a good understanding nowadays that OHC amplify vibrations of the BM within a limited frequency range simply because timing of the OHC forces is optimal within this frequency range. There is, however, no restriction on OHC excitation only within this limited frequency range. OHCs can well be excited over much broader frequency range but they are not able to undamp BM responses over this much broader range. Cited Ren et al. (2016) is a good illustration of this concept.

We disagree with this criticism. In our opinion, there is no "good understanding" at present – there is merely widespread (but certainly not universal) acceptance of a hypothetical proposal that OHCs "amplify" BM motion, and that timing must be important if they do. We believe that alternative hypotheses may still be worthy of consideration, including those whereby OHC's simply control the energy flow within the cochlea (perhaps without the need for precise timing and/or amplification). We have tried to portray this belief more clearly and constructively in the revised manuscript.

Page 13. "BM motion is smaller than hotspot motion, and the 0.25-cycle BM/hotspot phase difference at low frequencies is inconsistent with the high speed of OHC length changes."

*But the OHCs are excited by relative displacement between the TM and RL **and any phase relationship could happen**. Also see our comment above.*

We are unsure how to respond to this point. The assertion that "any phase relationship could happen" does not seem consistent with reviewer #3's view, expressed earlier, that the "timing of the OHC forces" is crucial to the operation of the cochlear amplifier.

We have generalized our claim to include the observation of hotspot-to-BM phase differences in excess of 0.25 cycles.

Page 14. Paragraph starting “We propose that hotspot motion plays an important functional role in enhancing cochlear frequency selectivity.”

*Provides quite vague description of what the authors think is happening during travelling wave propagation. **What do you mean by “to become more focused than it is in the dead organ”?***

We meant two things, one of which is conventional wisdom, and the second of which is relatively new: (i) Sound-evoked motion is more focused along the length of the cochlear partition in a healthy cochlea (this has been shown by numerous investigators), and (ii) sound-evoked motion is more focused (into the DC/OHC hotspot region) within each (transverse) cross-section of the partition when the cochlea is alive (one example of this is illustrated in Fig. 4).

We have amended the text to be more specific and restrict ourselves to meaning (ii) now. In the revised MS, we specify: “...causing it to become more tightly focused within the cross-section of the organ of Corti than it would be otherwise be (e.g. in the dead organ, cf. Fig. 4B).”

Page 14. “On the other hand, reports on distortion product emissions have provided indirect evidence of an origin of distortion products remote from the CF region, at a more basal site (45).”

These results are readily explained by a shallow high-frequency slope of suppression tuning curves (e.g. see Nam and Guinan (2018)).

In the interests of brevity, we consider this to be beyond the scope of the current paper, but hope that our findings might be followed up by the otoacoustic emission research community.

We have toned down the offending paragraph to appear less opinionated (using “may also” / “may help to” / “could call for”, etc.).

Page 14. Paragraph starting “The wideband compression observed in the hotspot is puzzling from a functional point of view.”

*and the next paragraph. **It is not puzzling at all if you separate the concepts of amplification and OHC excitation** as indicated in our comment above. The authors have to accept later in the same paragraph that “BM motion represents the “true” carrier of the low-frequency wave” but **they are still too shy to accept that** amplification of the BM responses and OHC excitation and resultant compression of responses within the organ of Corti are different phenomena.*

We are unsure how to respond to these comments. We have tried to keep these concepts completely separate in our MS, and have only referred to “amplification” as a hypothetical concept (whereas OHC excitation is undoubtedly real). We have also been careful to refer to our own observations as showing compression (not “amplification”), and it is the wideband nature of this phenomenon which we thought puzzling: we suspect that the low-frequency compression that we observe in the hotspot region is an epiphenomenon (and discuss it as such), in much the same way that reviewer #3 may suspect OHC forces to be when their timing is not “optimal” to “amplify vibrations of the BM”. What we are currently unwilling (but definitely not “too shy”) to accept is that cochlear compression and amplification are either intimately or functionally related.

*Last three paragraphs of the Discussion. The idea of parametric control of the OHC operation/impedance matching is not novel and the authors should **provide references to relevant publications.***

We do not claim that this idea is new, and already cited one relevant previous publication (reference 10, published in 1999).

We have added a second, even older reference (to Allen, 1990), and we are running out of space for more.

We have amended the text to clarify that “Such a proposal is by no means new (see ^{58,11}), but has been overshadowed for many years by the idea that there is a real “cochlear amplifier”, involving the cycle by cycle injection of extrinsic energy into the BMs vibrations. The functional role of a parametric impedance regulation is to compress the local IHC input, rather than to amplify BM motion, and the “functional part” of the compression is restricted to a narrow frequency band around CF (as reflected by BM nonlinearity).”

REVIEWERS' COMMENTS:

Reviewer #1 (Remarks to the Author):

Very nice paper!

Reviewer #2 (Remarks to the Author):

The authors have satisfactorily addressed my comments and suggestions from the previous round of review. The clarity of the manuscript has been improved significantly as a result of the reviewers' comments, particularly those from reviewer #3.

Authors responses (**bold**) to reviewers' comments (italic):

Reviewer #1 (Remarks to the Author):

Very nice paper!

We thank reviewer #1 for this praise. We do not disagree 😊

Reviewer #2 (Remarks to the Author):

The authors have satisfactorily addressed my comments and suggestions from the previous round of review. The clarity of the manuscript has been improved significantly as a result of the reviewers' comments, particularly those from reviewer #3.

We thank reviewer #2 for these supportive and well timed comments.